# Kinetics of cytokine receptor trafficking determine signaling and functional selectivity

Jonathan Martinez-Fabregas[1‡], Stephan Wilmes[1‡], Luopin Wang[2‡], Maximillian Hafer[3], Elizabeth Pohler[1], Juliane Lokau[4], Christoph Garbers[4], Adeline Cozzani[5], Paul K Fyfe[1], Jacob Piehler[3], Majid Kazemian[2†*], Suman Mitra[5†*], Ignacio Moraga[1†*]

[1]Division of Cell Signaling and Immunology, School of Life Sciences, University of Dundee, Dundee, United Kingdom; [2]Department Computer Science, Purdue University, West Lafayette, United States; [3]Department of Biology, University of Osnabrück, Osnabrück, Germany; [4]Department of Pathology, Medical Faculty, Otto-von-Guericke-University Magdeburg, Magdeburg, Germany; [5]INSERM UMR-S-11721, Centre de Recherche Jean-Pierre Aubert (JPARC), Institut pour la Recherche sur le Cancer de Lille (IRCL), Université de Lille, Lille, France

*For correspondence:
kazemian@purdue.edu (MK);
suman.mitra@inserm.fr (SM);
IMoragagonzalez@dundee.ac.uk (IM)

[†]These authors contributed equally to this work
[‡]These authors also contributed equally to this work

Competing interests: The authors declare that no competing interests exist.

**Abstract** Cytokines activate signaling via assembly of cell surface receptors, but it is unclear whether modulation of cytokine-receptor binding parameters can modify biological outcomes. We have engineered IL-6 variants with different affinities to gp130 to investigate how cytokine receptor binding dwell-times influence functional selectivity. Engineered IL-6 variants showed a range of signaling amplitudes and induced biased signaling, with changes in receptor binding dwell-times affecting more profoundly STAT1 than STAT3 phosphorylation. We show that this differential signaling arises from defective translocation of ligand-gp130 complexes to the endosomal compartment and competitive STAT1/STAT3 binding to phospho-tyrosines in gp130, and results in unique patterns of STAT3 binding to chromatin. This leads to a graded gene expression response and differences in ex vivo differentiation of Th17, Th1 and Treg cells. These results provide a molecular understanding of signaling biased by cytokine receptors, and demonstrate that manipulation of signaling thresholds is a useful strategy to decouple cytokine functional pleiotropy.

## Introduction

Cytokines modulate the immune response by activating a common JAK/STAT signaling pathway upon cell surface receptor dimerization/oligomerization (*Gorby et al., 2018*; *Stroud and Wells, 2004*; *Wang et al., 2009*; *Wells et al., 1993*). A conundrum in the field pertains to how biological specificity is achieved in the cytokine system by using such reduced number of signaling intermediaries, that is four JAKs and seven STATs (*Murray, 2007*; *Schindler et al., 2007*). Indeed, there are numerous examples in the literature where cytokines activating the same STATs in CD4 T cells, for example IL-6 and IL-10 (*Grotzinger et al., 1997*; *Walter, 2004*), produce opposite responses, that is pro-inflammatory vs anti-inflammatory responses respectively (*Hunter and Jones, 2015*; *Wilson et al., 2005*).

In recent years, a number of studies in multiple cytokine systems have shown that cytokine signaling is not an 'all or none' phenomenon and can be modulated by alterations of cytokine-receptor binding properties (*Spangler et al., 2015*). Changes in cytokine-receptor binding kinetics and strength were shown to play a crucial role in defining type I and type III interferons biological

potencies (*Mendoza et al., 2017*; *Pestka, 2007*; *Piehler et al., 2012*; *Subramaniam et al., 1995*). A mutation in erythropoietin (Epo) found in humans, which reduces its binding affinity for its receptor (EpoR), was shown to bias signaling output by EpoR and caused severe anemia in human patients (*Kim et al., 2017*). Biased EpoR signaling was also achieved using surrogate Epo ligands that altered the receptor binding topology (*Moraga et al., 2015b*). Cross-reactive cytokine-receptor systems, where shared receptors engage multiple cytokines and elicit differential responses is another example where receptor binding properties influence signaling and activity, for example the IL-4/IL-13 system (*Heller et al., 2008*; *LaPorte et al., 2008*), IL-2/IL-15 system (*Ring et al., 2012*; *Rochman et al., 2009*; *Waldmann, 2006*) and the IL-6 family system (*Wang et al., 2009*). Viruses often encode for cytokine-like proteins that bind cytokine receptors with altered binding properties, providing them with means to fine-tune the immune response to their own advantage (*Boulanger et al., 2004*; *Walter, 2004*). All these examples strongly argue in favour of cytokine-receptor binding parameters contributing to regulate signaling, however a model providing molecular bases for signaling biased by cytokine receptor is missing.

Biased signaling is not a unique feature of the cytokine family. G-protein-coupled receptors (GPCRs), which contain seven-transmembrane domains, are the classical system where biased signaling was first described (*Hilger et al., 2018*; *Wootten et al., 2018*). In this system, different ligands binding a common receptor can trigger differential signaling programs by instructing specific allosteric changes in the transmembrane α-helices of the receptor (*Hilger et al., 2018*; *Wootten et al., 2018*). However, this mechanism is more difficult to imagine for cytokine receptors where the transmembrane (TM) region contributes less significantly to signaling. Cytokine receptor chimeras where their TM has been swapped by that of other receptors still trigger signaling (*Sharma et al., 2016*). How can cytokine receptors trigger biased signaling responses then? A common feature to all cytokine systems is that upon ligand stimulation cytokine-receptor complexes traffic to intracellular compartments, where they are often degraded, contributing to switching off the response (*Becker et al., 2010*; *Bulut et al., 2011*; *Claudinon et al., 2007*; *German et al., 2011*; *Keeler et al., 2007*; *Shah et al., 2006*). However, a complex positive regulatory role of endocytosis in cytokine signaling has emerged (*Becker et al., 2010*; *Cendrowski et al., 2016*; *Fallon and Lauffenburger, 2000*; *Marchetti et al., 2006*; *Sarkar et al., 2002*). Several studies recently suggested a novel role for the endosomal compartment in stabilizing cytokine receptor dimers by enhancing local receptor concentrations (*Gandhi et al., 2014*; *Moraga et al., 2015a*), thus contributing to signaling fitness even at low complex stabilities. In agreement with this model, mutations on cytokine receptors that alter their intracellular traffic can result in activation of novel or deregulated signaling programs causing disease (*Reddy et al., 1996*). Furthermore, activated JAK/STAT proteins have been described in endosomes after interferon stimulation, suggesting that signaling continues upon receptor internalization (*Payelle-Brogard and Pellegrini, 2010*). How changes in cytokine-receptor complex half-life and endosomal trafficking fine-tunes cytokine signaling and biological responses requires further investigation.

Here, using model cell lines and primary human CD4 T cells, we systematically explored how modulation of cytokine-receptor complex stability impacts signaling identity and biological responses, using IL-6 as a model system. IL-6 is a highly pleiotropic cytokine, which critically contributes to mounting the inflammatory response (*Grotzinger et al., 1997*; *Hunter and Jones, 2015*; *Naka et al., 2002*). IL-6 stimulation drives differentiation of Th17 cells (*Jones et al., 2010*; *Kimura and Kishimoto, 2010*; *Louten et al., 2009*), and inhibits the differentiation of Th1 (*Diehl and Rincón, 2002*) and T regulatory (reg) cells (*Kimura and Kishimoto, 2010*; *Korn et al., 2008*). Deregulation of IL-6 levels and activities is often found in human diseases, making IL-6 a very attractive therapeutic target (*Hunter and Jones, 2015*). IL-6 exerts its immuno-modulatory activities by engaging a hexameric complex comprised of two molecules of IL-6Rα, two molecules of gp130 and two molecules of IL-6, leading to the downstream activation of STAT1 and STAT3 transcription factors (*Wang et al., 2009*). Using the yeast-surface display engineering platform, we isolated a series of IL-6 variants binding gp130 with different affinities, ranging from wild-type binding affinity to more than 2000-fold enhanced binding. Quantitative signaling and imaging studies revealed that reduction in cytokine-receptor complex stability resulted in differential cytokine-receptor complex dynamics, which ultimately led to activation of biased signaling programs. Low affinity IL-6 variants failed to translocate to intracellular compartments and induce gp130 degradation, triggering STAT3 biased responses. Indeed, inhibition of gp130 intracellular translocation by chemical or genetic

blockage of clathrin-mediated trafficking, reduced STAT1 activation levels, without affecting STAT3 activation. Through a series of molecular and cellular assays we demonstrated that STAT1 requires a higher number of phospho-Tyr available in gp130 to reach maximal activation, explaining its enhanced sensitivity to changes in cytokine-receptor dwell-times. The biased signaling programs engaged by the IL-6 variants did not have a linear effect on STAT3 transcriptional activities. Reduced STAT3 activation levels by the low affinity IL-6 variants resulted in graded STAT3 binding to chromatin and gene expression, with some genes exhibiting a high degree of sensitivity to STAT3 activation levels, and other genes being equally induced by all three IL-6 variants. Moreover, IL-6 immunomodulatory activities exhibited different sensitivity thresholds to changes on STAT activation levels, with Th17 differentiation being induced by all three variants, and inhibition of Treg and Th1 differentiation only robustly promoted by the high affinity variant. Our results provide a molecular model using spatio-temporal dynamics of cytokine-receptor complexes and competitive binding of STATs proteins for phosphorylated tyrosine residues, to explain how cells integrate cytokine signaling signatures into specific biological responses through the establishment of different gene induction thresholds. At the more practical level, our results highlight that manipulation of cytokine-receptor binding parameters via protein engineering is a useful strategy to decouple cytokine functional pleiotropy, a major source of unwanted side effects and toxicity in cytokine-based therapies.

## Results

### Engineering IL-6 variants with different binding affinities for gp130

IL-6 triggers signaling by assembling a hexameric complex probably in a three steps process (*Figure 1a*): IL-6 first binds to the IL-6Rα receptor subunit with nanomolar affinity via its site-1 binding interface. In the second step, the IL-6/IL-6Rα complex recruits one molecule of gp130 via IL-6 site-2 binding interface to form a hetero-trimeric complex. Finally, two of these hetero-trimeric complexes dimerize via the IL-6 site-3 binding sites to form a signalling-active hexameric complex (*Figure 1a*, top panel) (*Wang et al., 2009*). Importantly, IL-6Rα further contributes to stabilize site-2 and site-3 interaction of IL-6 by directly interacting with gp130 (*Figure 1—figure supplement 1a*). In an inflammatory environment, IL-6Rα is shed from the plasma membrane via proteases of the ADAM family, and binds IL-6 with high affinity to form a soluble stable complex. This complex triggers potent signaling by recruiting two molecules of gp130 via site-2 and site-3 IL-6 binding interfaces therefore contributing to enhance the inflammatory response (*Figure 1a*, middle panel) (*Wang et al., 2009*). This inflammatory complex can be mimicked by a linker-connected single-chain variant of soluble IL-6Rα and IL-6, called hyper-IL-6 (HyIL-6) (*Fischer et al., 1997*).

Here we ask whether modulation of IL-6-gp130 binding parameters would instruct different signaling outcomes and decouple IL-6 functional pleiotropy. To address this question, we have used yeast surface display to engineer a series of IL-6 variants binding gp130 with different affinities, therefore providing different degrees of IL-6Rα dependency (*Figure 1a* bottom panel). As described above, IL-6 interacts with gp130 on two sites, named site-2, which uses helixes A and C on IL-6, and site-3 which uses part of the AB-loop and helix D (*Figure 1b*) (*Wang et al., 2009*). We focused on the site-2 binding interface because this interface seems to be the main driver of gp130-IL6 interaction in the absence of IL-6Rα. Using the existing crystal structure of the IL-6 hexameric complex, we identified 14 amino acids on IL-6 forming the site-2 binding interface, which we randomized using a 'NDT' degenerate codon encoding amino acids: G,V,L,I,C,S,R,H,D,N,F,Y (*Figure 1b*). The resulting library contained more than $3 \times 10^8$ unique variants.

The library was selected for gp130 binders through five rounds of selection in which the gp130 concentration was gradually decreased from 1 μM to 1 nM (*Figure 1c–e*). Nine clones were selected based on their on-yeast binding titrations and their sequences were obtained (*Figure 1—figure supplement 1b*). From this initial library, IL-6 variants exhibiting a wide range of binding affinities for gp130 were isolated, ranging from wild type (wt) affinity (A1) to 200-fold better binding (F3) (*Figure 1f*). In order to isolate IL-6 variants binding with even higher affinity to gp130, we performed a second library, where we further engineered the F3 mutant, by carrying out a soft randomization of the amino acids forming the gp130 site-2 binding interface. After five additional rounds of selection we isolated three new variants (Mut1, Mut3 and Mut7) that bound gp130 with an apparent on-yeast binding $K_D$ of 2 nM (*Figure 1f*). To perform surface plasmon resonance (SPR) studies, we

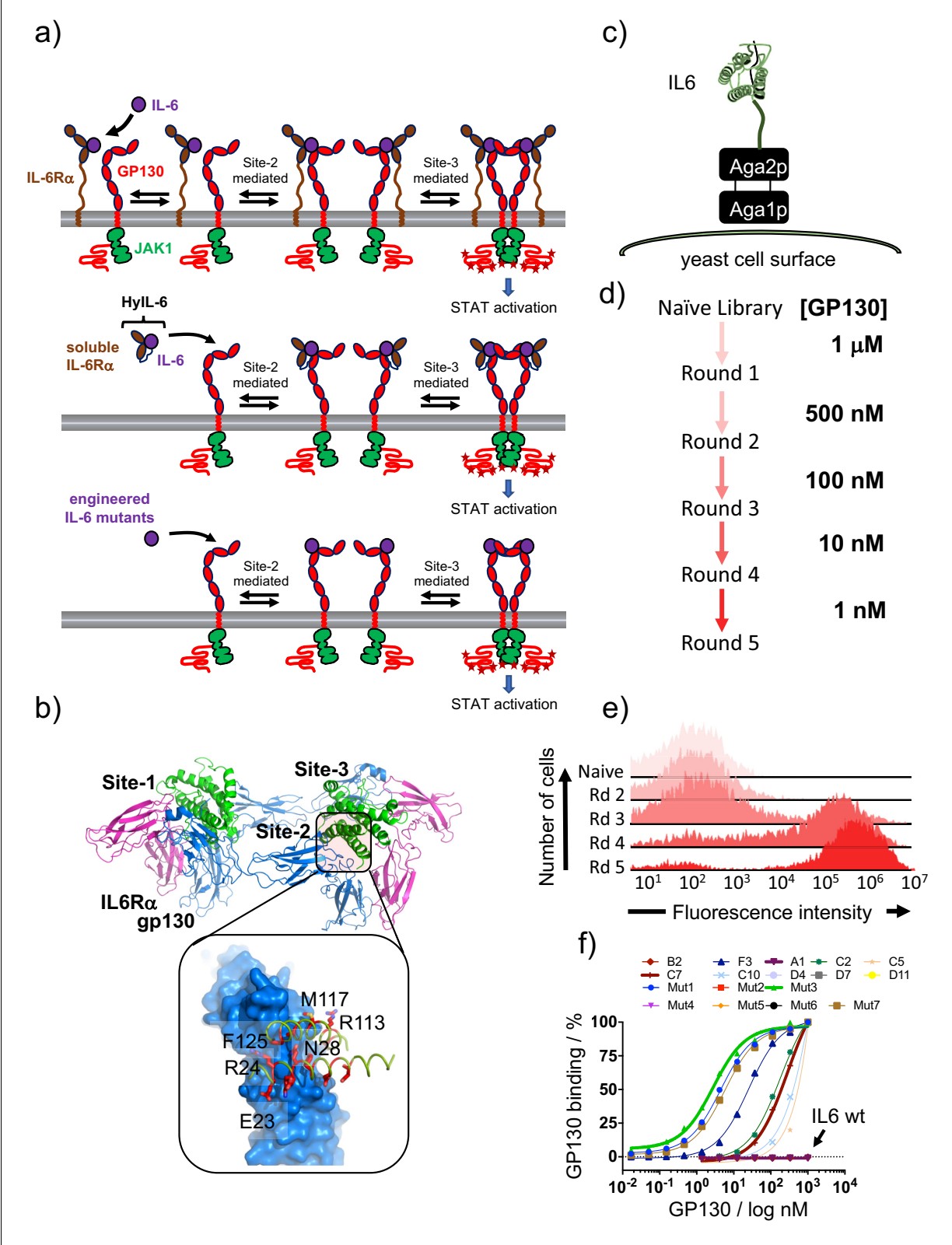

**Figure 1.** Isolation of IL-6 variants binding gp130 with different affinities. (**a**) Schematic representation of IL-6 receptor assembly kinetics elicited by IL-6 wildtype (top panel), HyIL-6 (middle panel) and IL-6 mutants molecules (bottom panel). (**b**) Crystal structure of IL-6, in green, bound to gp130 and IL-6Rα ectodomains, in blue and pink respectively. Inlet highlights the IL-6/gp130 site-2 binding interface. Amino acids included in the library design are colored in red. (**c**) Schematic representation of IL-6 display in the yeast surface via aga2p-aga1p interaction. (**d**) Work-flow of IL-6 library selection

*Figure 1 continued on next page*

Figure 1 continued

process. Five rounds of selection were undertaken, starting with 1 μM of gp130 ectodomain and finishing with 1 nM. (e) Representative gp130 staining of the selected IL-6 library. The five rounds of selections were incubated with 1 μM of biotinylated gp130 for 1 hr followed by 15 min incubation with SA-alexa647. Early rounds exhibit weak binding to gp130, but as the library converged into few high affinity clones, the gp130 staining improve significantly. (f) Dose/Response gp130 binding curves performed in single yeast colonies, each encoding a different IL-6 variant. Gp130 concentration started at 1 μM, and eight different concentrations in a 1/3 dilution series were tested.

The online version of this article includes the following figure supplement(s) for figure 1:

**Figure supplement 1.** Biophysical characterization of IL-6 variants.

recombinantly expressed all these mutants and purified them to homogeneity. These studies confirm the same trend observed in the on-yeast binding titration experiments, with values ranging from 648 nM (A1) to 6.2 nM (C7) and 379 pM (Mut3) (*Figure 1—figure supplement 1c-d*). While we could not detect binding of gp130 to A1 and IL-6 wt in the on-yeast binding titration studies, we determined accurate $K_D$ binding constants in the SPR studies. It is important to emphasize that our engineered IL-6 variants still exhibit wt binding affinity for gp130 via their site-3 interfaces. Thus, although they bind gp130 in the absence of IL-6Rα, the overall half-life of the tetrameric surface complex formed by these variants is defined by their combined site-2/site-3 binding interaction. HyIL-6, via additional site-2/site-3 stabilization resulting from IL-6Rα/gp130 contacts, forms an even longer-lived complex than any of the engineered IL-6 variants.

## IL-6 variants induce differential STAT3/STAT1 activation ratios

IL-6 binding to gp130 triggers the phosphorylation and activation of STAT1 and STAT3 effector proteins via JAK1, which largely determine cellular responses. We therefore studied the different STAT1 and STAT3 activation signatures elicited by the A1, C7 and Mut3 IL-6 variants. For that, we used HeLa cells, which express very low levels of IL-6Rα subunit and therefore allow us to study the contribution of gp130 binding to signaling output by the IL-6 variants. As control we used IL-6 wt, which requires IL-6Rα expression to activate signaling, and Hyper IL-6 (HyIL-6), which binds gp130 with high affinity and potently triggers signaling in cells lacking IL-6Rα (*Fischer et al., 1997*). The three engineered variants exhibited different degrees of IL-6Rα dependency based on their gp130 binding affinities (*Figure 2—figure supplement 1a-b*). As expected, while IL-6 wt stimulation led to a poor signaling response in HeLa cells, HyIL-6 stimulation produced a robust STAT1 and STAT3 activation in dose-response studies (*Figure 2a–b*). Interestingly, different engineered IL-6 variants drove differential phosphorylation amplitudes in STAT1 and STAT3 (*Figure 2a–b*). These differences in signaling amplitudes could not be rescued by further increases in ligand concentration (*Figure 2a–b*), nor were the result of altered signaling kinetics induced by the IL-6 variants (*Figure 2c–d*). Strikingly, we observed that STAT1 phosphorylation was profoundly more affected than STAT3 by changes in gp130 binding affinities (*Figure 2a–d*). While Mut3 activated STAT1 and STAT3 to the same extent than HyIL-6, the C7 variant induced 70% of the STAT3 phosphorylation levels but only 25% of the STAT1 phosphorylation levels induced by HyIL-6. Similarly, the A1 variant induced 50% of the STAT3 phosphorylation levels as compared to HyIL-6, but failed to induce STAT1 phosphorylation (*Figure 2a–d*). This biased STAT3 activation by the IL-6 variants resulted in altered STAT3/STAT1 activation ratios, with IL-6 variants binding with lower affinity to gp130 exhibiting a disproportionally high activation of STAT3 versus STAT1 (*Figure 2e* and *Figure 2—figure supplement 1c*).

To investigate whether the biased STAT3 signature induced by the different IL-6 variants would impact their transcriptional programs, we analysed the induction of a classical STAT1-dependent and STAT3-dependent proteins, that is IRF1 and ICAM-1, by the three IL-6 variants (*Gil et al., 2001*; *Wung et al., 2005*). HeLa cells were stimulated with saturating concentrations of the different variants and the levels of IRF1 and ICAM1 expression were measured by flow cytometry. As shown in *Figure 2f*, induction of IRF1 expression was more sensitive to changes on gp130 binding affinity, paralleling the sensitivity of STAT1 activation. Overall these data indicate that modulation of cytokine-receptor binding parameters decouples signaling output and transcriptional programs.

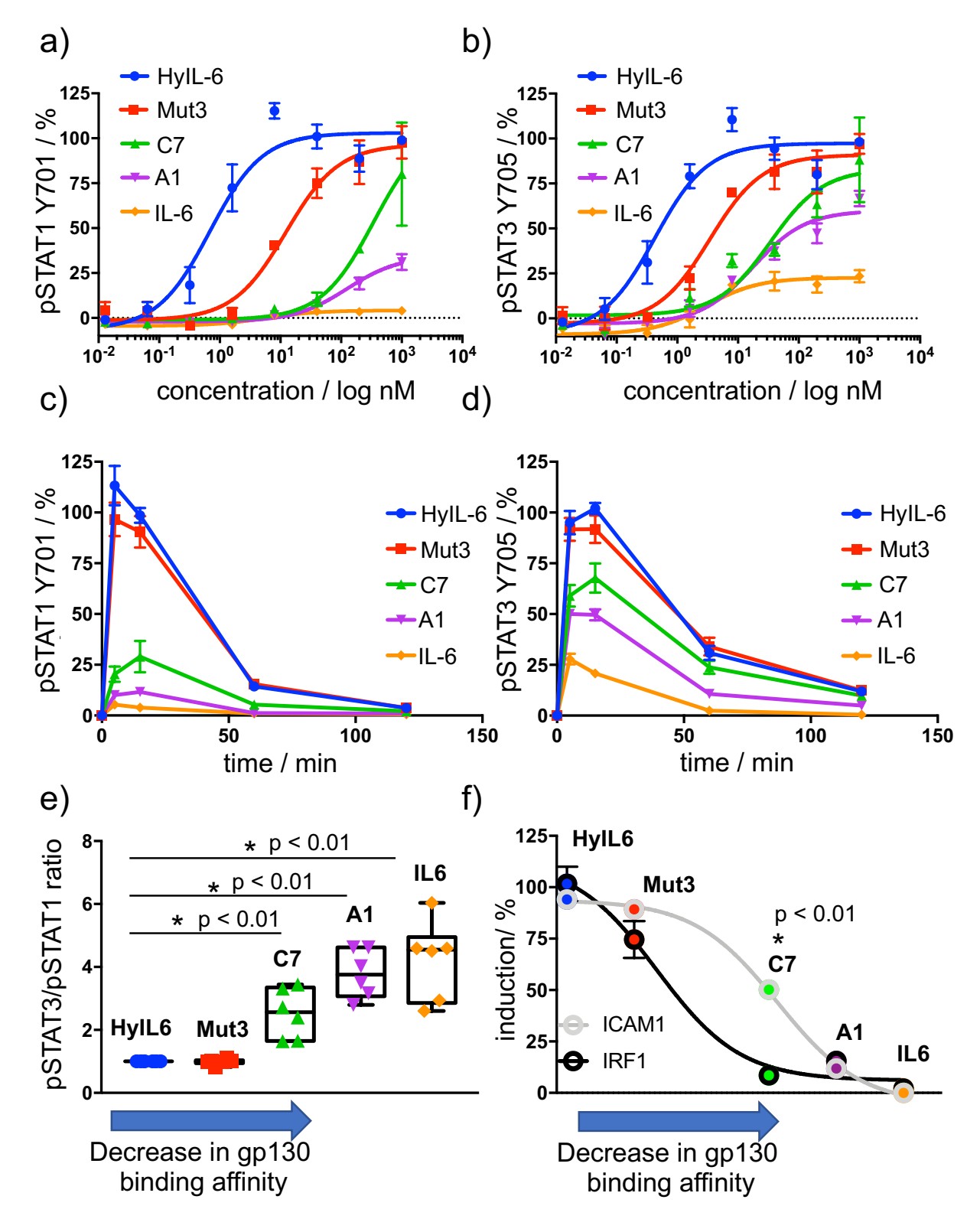

**Figure 2.** Determination of signaling signatures activated by IL-6 variants. (a–b) HeLa cells were stimulated with the indicated doses of IL-6 ligands for 15 min and levels of STAT1 (a) and STAT3 (b) were analyzed by phospho-flow cytometry. Sigmoidal curves were fitted with GraphPath Prism software. Data are mean + /- SEM from three independent replicates, each performed in duplicate. (c–d) HeLa cells were stimulated with 100 nM of IL-6 ligands for the indicated times and the levels of STAT1 (c) and STAT3 (d) were analyzed by phospho-Flow cytometry. Data are mean + /- SEM from three

*Figure 2 continued on next page*

*Figure 2 continued*

independent replicates, each performed in duplicate. (e) Differential STAT activation by engineered IL-6 ligands. pSTAT3/pSTAT1 ratios are plotted for all the IL-6 ligands. An arrow indicating the binding affinity trends of each ligand was placed in the X axis of the plot. Low gp130 affinity ligands exhibit a more pronounced STAT3/STAT1 ratio than high affinity ligands. Data are mean + /- SEM from three independent replicates, each performed in duplicate. (f) Comparison of STAT1- (IRF1) and STAT3- dependent (ICAM1) gene induction by engineered IL-6 ligands. HeLa cells were stimulated with saturating concentrations (100 nM) of the different IL-6 ligands for either 2 hr (IRF1) or 24 hr (ICAM1) and the levels of IRF1 and ICAM1 induction were measured via flow cytometry. Data are mean + /- SEM from three independent replicates, each performed in duplicate. An arrow indicating the binding affinity trends of each ligand was placed in the X axis of the plot.

The online version of this article includes the following figure supplement(s) for figure 2:

**Figure supplement 1.** IL-6Rα dependency of the IL-6 variants.

## Short-lived IL-6-gp130 complexes fail to traffic to intracellular compartments

We have shown that engineering IL-6 to display different binding dwell-times for gp130 results in biased STAT3 and STAT1 responses by this receptor system. However, the molecular basis that allow fine-tuning of gp130 signaling outputs in response to changes in ligand-receptor complex half-life remains unclear. To gain molecular insight into this question, we probed the assembly of gp130 dimers at the single molecule level using dual-colour total internal reflection fluorescence (TIRF) microscopy. In these experiments, we transfected genome-engineered RPE1 cells lacking endogenous gp130 expression (*Figure 3—figure supplement 1a–b*) with gp130 N-terminally tagged with a meGFP, which was rendered non-fluorescent by the Y67F mutation (*Figure 3a*). This tag (mXFP) is recognized by dye-conjugated anti-GFP nanobodies (NB), allowing quantitative fluorescence labeling of gp130 at the cell surface of live cells (*Figure 3—figure supplement 1c*). Well-balanced dual-colour labeling was achieved using equal concentrations of nanobodies either conjugated to RHO11 or DY647, allowing us to probe diffusion and interaction of individual receptor molecules in the plasma membrane of live cells as recently shown in *Kim et al. (2017)*, and *Moraga et al. (2015b)*. Single-molecule co-localization and co-tracking analysis was used to identify correlated motion (co-locomotion) of the two spectrally separable fluorophores, which was taken as readout for gp130 dimerization. In absence of ligand stimulation, no significant gp130 dimer levels were observed. After cytokine stimulation, strong gp130 dimerization was found for HyIL-6 and Mut3, in agreement with the model of ligand-induced cytokine receptor assembly. Significantly lower dimerization levels were found for C7 whereas A1 and IL-6 wt did not yield dimer levels above background (*Figure 3b*, *Figure 3—figure supplement 1c*, *Figure 3—figure supplement 2a*). In line with the dimerization, we observed a considerable decrease in lateral diffusion mobility, which can be ascribed to increased friction of dimeric receptors within the membrane (*Moraga et al., 2015b*; *Wilmes et al., 2015*) (*Figure 3—figure supplement 2b-c*). Bleaching experiments at elevated laser intensities confirmed formation of receptor homodimers (*Figure 3c* and *Figure 3—figure supplement 2d*).

Interestingly, although we detected strong STAT3 activation by C7 and A1 variants (*Figure 2a–d*), their ability to dimerize gp130 in live cells was significantly compromised (*Figure 3b*). Based on this, we speculated that complexes formed by these variants were too short-lived and escaped detection by single molecule tracking. It is accepted that cytokine-receptor complex rapidly traffic to intracellular compartments, where they can be degraded or recycled (*Gonnord et al., 2012*). Recently, it has been proposed that endosomes could act as signaling hubs, helping to sustain low-affinity cytokine-receptor dimers by enhancing the local receptor density (*Gandhi et al., 2014*). Thus, we asked whether the biased signaling program engaged by the three IL-6 variants resulted from differential receptor trafficking. To test this hypothesis, we fluorescently labeled HyIL-6 and the three IL-6 mutants and followed their receptor-mediated internalization by confocal imaging. Importantly, the dye-conjugated IL-6 variants induced dimerization of endogenous gp130 in HeLa cells, confirming their functionality (*Figure 3—figure supplement 2e*). HeLa cells were incubated for 30 min with the labeled cytokines and their internalization monitored by confocal microscopy. Anti-EEA1 antibodies were used to label early endosomes. As shown in *Figure 3d–e*, we detected high levels of labeled HyIL-6 in intracellular compartments that partially co-localized with EEA1 early endosome marker. Yet, much weaker levels of labeled Mut3 were detected, and no fluorescence was detected for C7 and A1 variants, despite moderate overexpression of gp130.

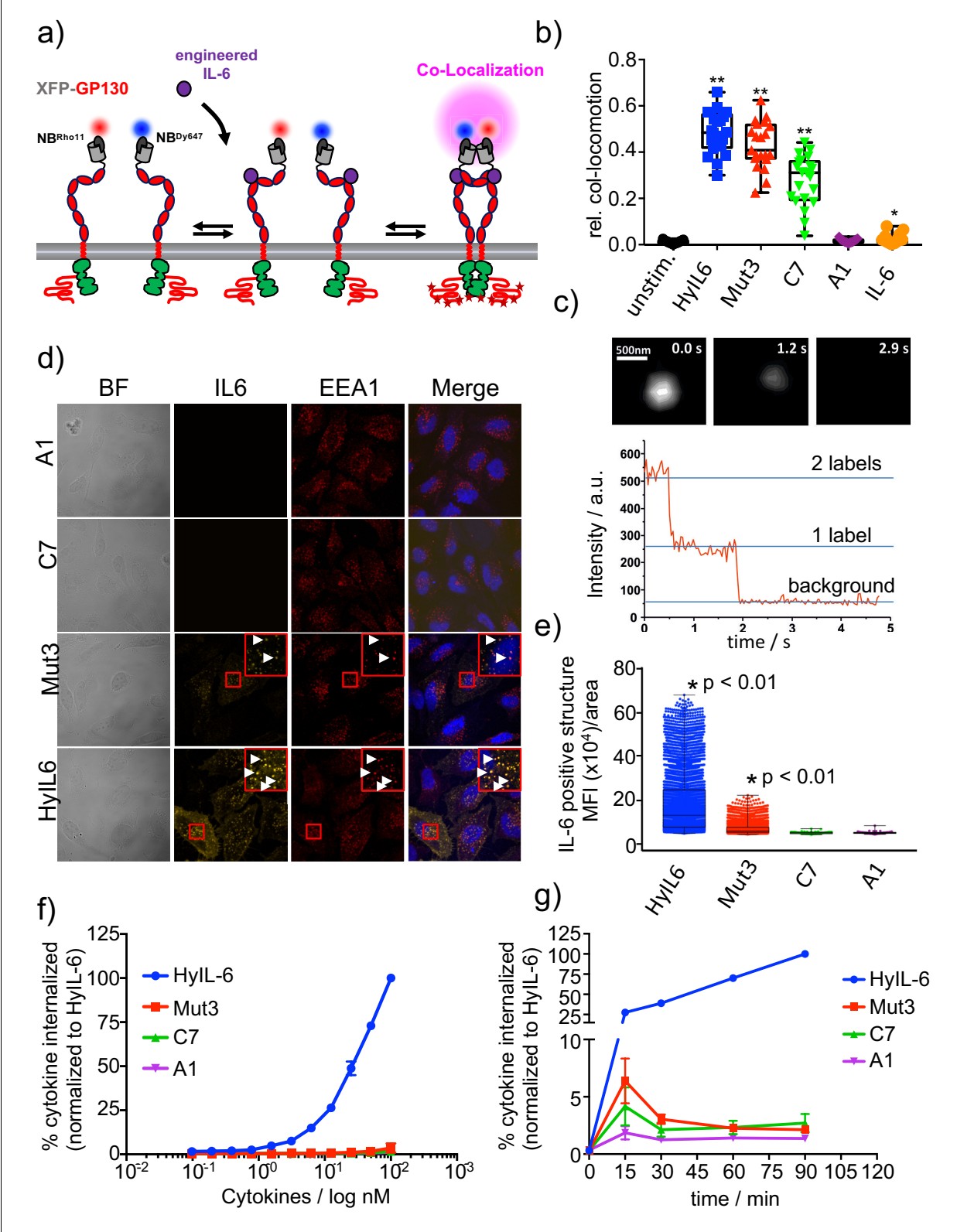

**Figure 3.** gp130 cell surface dimerization induced by the different IL-6 variants. (**a**) Quantification of gp130 homodimerization in the plasma membrane by dual-color single-molecule co-localization/co-tracking. mXFP-gp130 was expressed in RPE1 gp130KO cells and labeled via anti-GFP nanobodies conjugated with RHO11 and DY647, respectively. (**b**) Relative amount of co-trajectories for unstimulated gp130 and after stimulation with HyIL-6 and IL-6 mutants (Mut3, C7, A1 and IL-6 wt). (**c**) Single-color dual-step bleaching of a HyIL-6 induced gp130 dimer. (**d**) Uptake of DY547-conjugated HyIL-6 and

*Figure 3 continued on next page*

*Figure 3 continued*

IL-6 mutants. HeLa cells, overexpressing gp130 were stimulated for 45 min with 40 nM of each cytokine. Ligand uptake into endosomal structures was co-localized with EEA1. Nuclei were stained with DAPI (blue), shown in the merged image. Co-localization of ligands with EEA1 endosomes are highlighted in the zoomed area. (e) Quantification of Ligand binding/uptake. Mean fluorescence intensity of DY547-conjugated IL-6 variants colocalizing with EEA1 positive structures quantified using the Volocity 3D Image Analysis software (PerkinElmer). (f–g) Quantification of ligand uptake by flow-cytometry. (f) HeLa cells transfected with meGFP-GP130 were incubated with various doses of DY647-conjugated IL-6 variants for 15 min. (g) Kinetics of ligand uptake were measured after incubation with 100 nM of each DY647-conjugated IL-6. For both experimental series, cell surface bound ligands were removed by trypsination prior to flow cytometry. Data was in all cases normalized to HyIL-6 signal intensity.

The online version of this article includes the following figure supplement(s) for figure 3:

**Figure supplement 1.** Dimerization of gp130 induced by IL-6 variants.

**Figure supplement 2.** Characterization of gp130 induced complexes.

To ensure that the defective internalization of the IL-6 variants was not the result of a dose or a kinetic effect, we performed dose/response and kinetics studies on HeLa cells overexpressing gp130 (*Figure 3f–g*). In short, HeLa cells were incubated with the indicated doses of ligands (*Figure 3f*) or times (*Figure 3g*) and internalization of the labeled ligands was monitored by flow cytometry. After each indicated experimental point, cells were incubated with trypsin for 10 min to remove surface bound ligands. As shown in *Figure 3f and g* only HyIL-6 induced a robust internalization which was sustained over time. Mut3, C7 and A1 variants led to a poor but detectable transient internalization which correlated with their binding affinity. Overall, these data show that short-lived IL-6-gp130 complexes exhibit a defective intracellular traffic, which ultimately could explain the biased signaling programs engaged by the IL-6 variants.

## Gp130 internalization blockages differentially controls STAT1 activation

IL-6 stimulation drives proteasomal degradation of gp130 (*Tanaka et al., 2008*). Next, we studied whether stimulation of HeLa cells with the three IL-6 variants produced different levels of gp130 degradation. HeLa cells were stimulated with saturating concentrations of HyIL-6 or the different IL-6 variants for the indicated times in the presence of cycloheximide (CHX) to prevent new protein synthesis. As shown in *Figure 4a–b*, HyIL-6 induced the strongest gp130 degradation, followed by Mut3, which induced significantly lower gp130 degradation than HyIL-6. C7, A1 and IL-6 treatment did not result in gp130 degradation beyond what was induced by the CHX treatment, which consequently led to some stabilization of gp130 in the cell surface (*Figure 4a–b*). These results suggest that activation of signaling pathways leading to receptor degradation can be decoupled from STAT1/3 activation by modulating cytokine-receptor complex half-life. Indeed, while Mut3 activates STAT1 and STAT3 to the same extent as HyIL-6, it induced substantially lower degradation of gp130 (*Figure 4a–b*).

To investigate whether the defective gp130 internalization/degradation induced by the IL-6 variants was at the basis of their biased signaling program, we blocked gp130 internalization by incubating HeLa cells with Pitstop, a well-known clathrin inhibitor. First, we assayed the efficacy of Pitstop on blocking ligand-induced gp130 internalization. For that, we monitored fluorescently labeled HyIL-6 internalization in HeLa cells treated with Pitstop by flow cytometry. As shown in *Figure 4c*, internalization of HyIL-6 was strongly inhibited by Pitstop treatment, confirming the ability of this inhibitor to block gp130 internalization.

We next measured STAT1 and STAT3 phosphorylation levels induced by Mut3, C7 and A1 variants in HeLa cells pre-incubated with Pitstop (*Figure 4c–e*). As shown in *Figure 4d*, the STAT3 phosphorylation levels induced by the three IL-6 variants did not change when gp130 internalization was blocked. STAT1 activation by Mut3 was not affected and A1 failed to activate STAT1 as shown in previous experiments (*Figure 4c*). STAT1 phosphorylation levels induced by the C7 variant on the other hand were significantly downregulated in the presence of Pitstop (*Figure 4c*), which ultimately led to a more pronounced STAT3/STAT1 activation ratio by this variant (*Figure 4e*). We could confirm these data by silencing clathrin in HeLa cells using siRNA (*Figure 4f–h*, *Figure 4—figure supplement 1a-b*). Clathrin silencing did not affect STAT3 activation (*Figure 4g*), but reduced activation of STAT1 by Mut3 and C7 variants (*Figure 4f*). Overall, these data indicate that translocation of gp130 complexes to intracellular compartments is an important requisite for STAT1, but not STAT3 activation by short-lived IL-6-gp130 complexes.

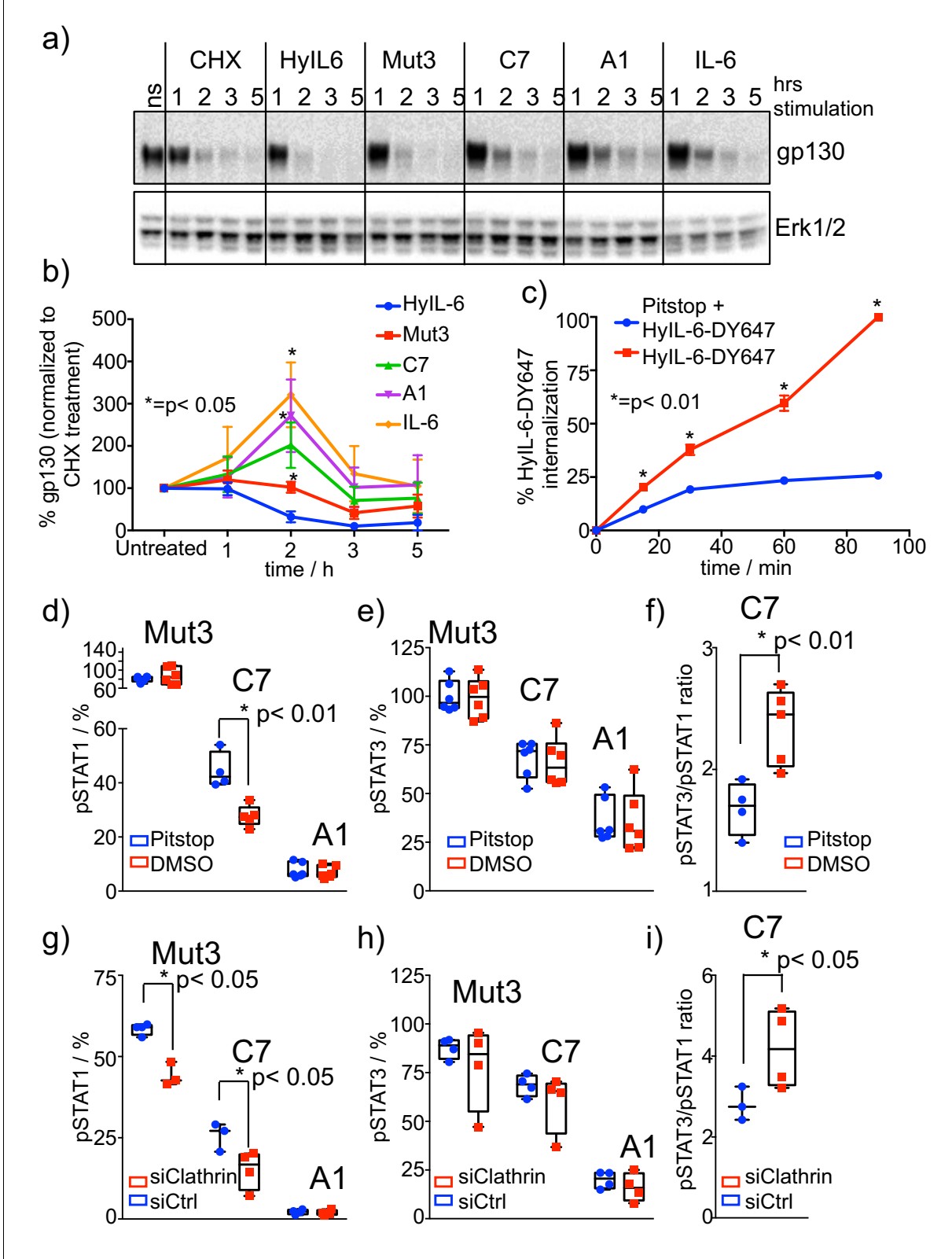

**Figure 4.** Role of receptor internalization in STAT activation by IL-6 variants. (a–b) HeLa cells were stimulated with saturating concentrations of the different IL-6 ligands for the indicated times in the presence of CHX to block new protein synthesis. Levels of gp130 were measured by western blotting using a gp130 specific antibody and quantified via ImageJ software. Values were normalized to the gp130 degradation levels induced by CHX alone. Data are mean + /- SEM of four independent experiments. (c) HeLa cells preincubated for 30 min with Pitstop or DMSO were incubated with

*Figure 4 continued on next page*

*Figure 4 continued*

fluorescently labeled HyIL-6 for the indicated times. HyIL-6 internalization was monitored by flow cytometry. Data are mean + /- SEM of three independent experiments. (**d–f**) HeLa cells preincubated for 30 min with Pitstop or DMSO were stimulated with saturating concentrations (100 nM) of the indicated IL-6 ligands for 15 min and levels of STAT1 (**d**) and STAT3 (**e**) activation were measured by phospho-Flow cytometry. Data are mean + /- SEM from three independent replicates, each performed in duplicate. The pSTAT3/pSTAT1 ratio calculated from these studies is plotted in (**f**). (**g–i**) HeLa cells were transfected with either control siRNA or clathrin specific siRNA. After 48 hours cells were stimulated with saturating concentrations (100 nM) of the indicated IL-6 ligands for 15 min and the levels of STAT1 (**g**) and STAT3 (**h**) activation were measured by phospho-Flow cytometry. Data are mean + /- SEM from two independent replicates, each performed in duplicate. The pSTAT3/pSTAT1 ratio calculated from these studies is plotted in (**i**). The online version of this article includes the following figure supplement(s) for figure 4:

**Figure supplement 1.** Clathrin silencing in HeLa cells.

## STAT1 and STAT3 compete for phospho-Tyrosines in the gp130 intracellular domain

We have shown that trafficking of IL-6/gp130 complexes to intracellular compartments preferentially modulates STAT1 activation. However, why STAT1 activation requires receptor internalization is not well defined. Previous work showed that STAT3, via its SH2 domain, binds with higher affinity to phospho-Tyr on gp130 than STAT1 (*Wiederkehr-Adam et al., 2003*). We thus postulated that competitive binding of STAT1 and STAT3 for phospho-Tyr on gp130 would result in differential levels of activation of these two transcription factors in the context of short-lived IL-6/gp130 complexes. To test this model, we generated a chimera receptor system, based on the IL-27 receptor complex, to study the influence of the number of phospho-Tyr available in gp130 on ligand-induced STAT1 and STAT3 activation.

IL-27 triggers signaling by dimerizing IL-27Rα and gp130 receptor subunits (*Stumhofer et al., 2010*). We took advantage of the shared use of gp130 by the two systems and swapped the intracellular domain of IL-27Rα with that of gp130. Additionally, we generated a second receptor chimera (gp130 ΔY), where the intracellular domain of IL-27Rα was swapped with the gp130 intracellular domain containing a deletion after JAK1 binding site, that is the box1-2 region. As a result of this, while the first chimera receptor can trigger the potential phosphorylation of eight Tyr, the second chimera can only induce the phosphorylation of four (*Figure 5a*). We then stably transfected all the constructs in RPE1 cells, which do not express IL-27Rα endogenously, but express endogenous levels of gp130 (*Figure 5b*). To ensure that all the RPE1 clones were homogenous and the effects that we see are specific, we compared the responsiveness of the three clones to HyIL-6. As shown in *Figure 5—figure supplement 1a*, HyIL-6 induced comparable levels of STAT1 and STAT3 activation in the three clones, strongly arguing that the endogenous gp130, JAK1, STAT1 and STAT3 levels in the three clones were identical. In response to IL-27, the three clones produced very similar STAT3 activation levels, suggesting that STAT3 activation is very efficient and only requires a minimal set of phospho-Tyr available to reach its activation peak (*Figure 5b* and *Figure 5—figure supplement 1b*). However, STAT1 activation levels dropped by more than fifty percent in the gp130 ΔY clone (*Figure 5b* and *Figure 5—figure supplement 1b*), demonstrating that STAT1 activation by gp130 requires a higher number of phospho-Tyr available.

We next studied the contribution of each independent Tyr found on the gp130 intracellular domain (ICD) to STAT1 and STAT3 activation. We focused on Tyr 767, 815, 905 and 915 because these Tyr have been reported to contribute the most to IL-6-induced signaling (*Gerhartz et al., 1996*; *Schmitz et al., 2000*). We generated seven different gp130 mutants with different pools of Tyr available for phosphorylation (*Figure 5c*) and transiently expressed them in gp130 KO HeLa cells (*Figure 5d*; *Figure 3—figure supplement 1a* and *Figure 5—figure supplement 1d*). We then quantified STAT1 and STAT3 phosphorylation levels induced by the indicated doses of HyIL-6 after 15 min stimulation. As shown in *Figure 5d*, single Tyr mutations did not decrease the STAT1/STAT3 activation levels induced by HyIL-6. Mutation of Tyr 905/915 and 815/905/915 to Phe produced a 50% decrease on the STAT1 phosphorylation levels induced by HyIL-6 stimulation, but only marginally affected STAT3 phosphorylation. Mutation of the four Tyr (Tyr767/815/905/915) to Phe resulted in almost complete loss of both STAT1 and STAT3 phosphorylation by HyIL-6. Interestingly, HyIL-6 triggered a biased STAT3 response in the double and triple gp130 Tyr mutants background. This data suggest that in non-optimal activation conditions, such as limited

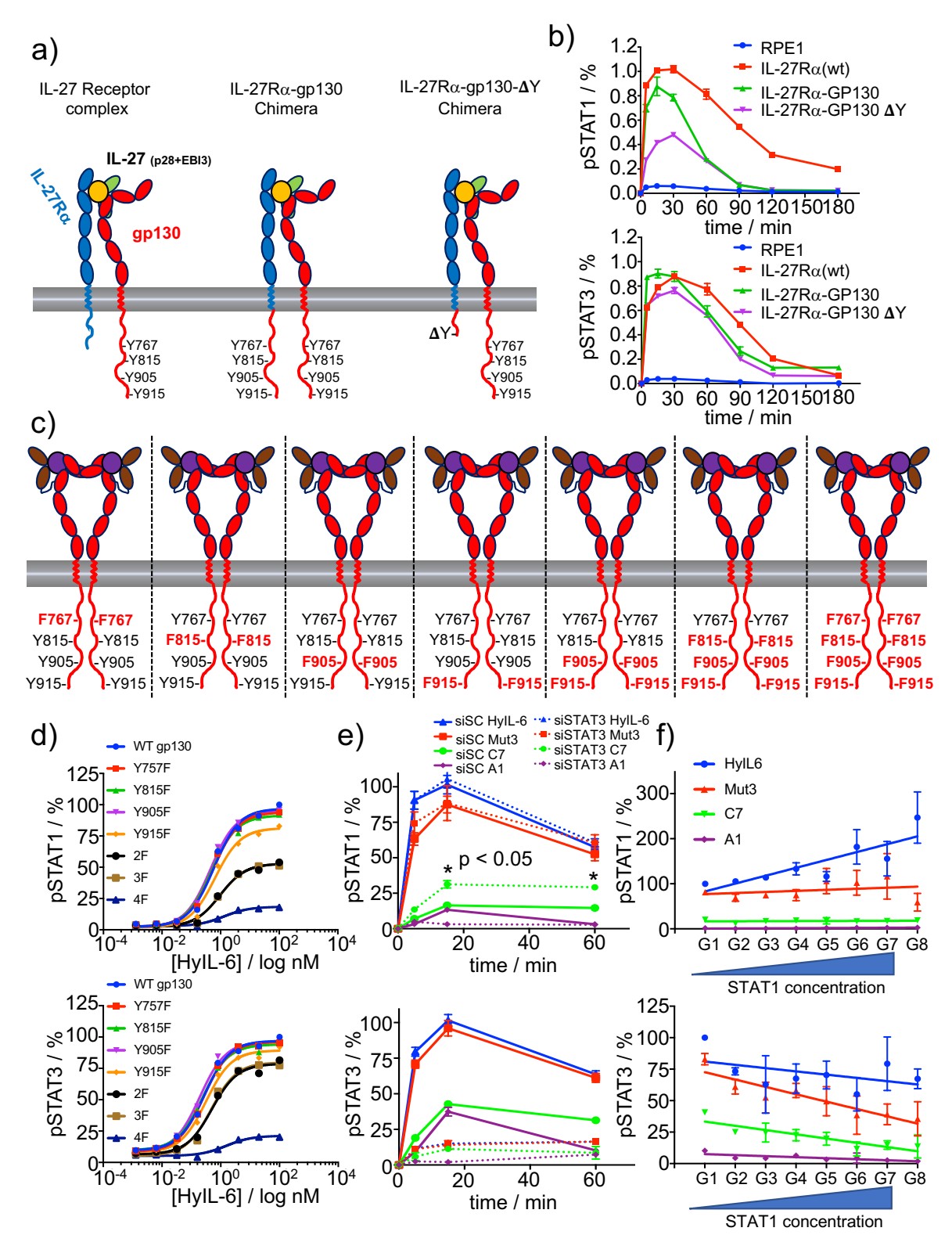

**Figure 5.** Correlation between number of P-Tyr in gp130 ICD and STATs activation. (a) Schematic representation of the different chimera receptors designed for this study. IL-27Rα intracellular domain was swapped for that of gp130 or a truncated version of the latter lacking all Tyr residues after the box1/2 region. This results in a receptor chimera complex able to engaged 8 P-Tyr and another one able to engage only 4 P-Tyr. RPE1 clones stably expressing the different receptor chimera constructs were generated. (b) Stable RPE1 clones were stimulated with saturating concentrations of IL-27 for

*Figure 5 continued on next page*

*Figure 5 continued*

the indicated times and the levels of STAT1 (left panel) and STAT3 (right panel) activation were measured by Phospho-Flow cytometry. Data are mean + /- SEM from three independent replicates, each performed in duplicate. (c) Schematic representation of seven different gp130 mutants, where the identity of the Tyr on gp130 ICD mutated to Phe is highlighted in red. (d) gp130 KO HeLa cells were transiently transfected with either gp130 wt or each of the gp130 mutants described in c and pSTAT1 (top panel) and pSTAT3 (bottom panel) levels upon stimulation with the indicated doses of HyIL-6 were monitored by flow cytometry. Data are mean + /- SEM from two independent experiments, each in duplicate. (e) Hela cells were transfected with either control siRNA or siRNA targeting STAT3. After 48 hr transfected cells were stimulated with saturating concentrations of the different IL-6 ligands for the indicated times and the levels of STAT1 (top panel) or STAT3 (bottom panel) activation were measured by Phospho-Flow cytometry. Data are mean + /- SEM from three independent replicates, each performed in duplicate. (f) HeLa cells were transiently transfected with STAT1-GFP and the levels of pSTAT1 and pSTAT3 upon stimulation with a saturated dose of HyIL-6 (20 nM) for 15 min were monitored by flow cytometry. G1-G8 represents gates on the flow cytometer denoting increasing STAT1-GFP expression levels. Data are mean + /- SEM from two independent replicates, each performed in duplicate.

The online version of this article includes the following figure supplement(s) for figure 5:

**Figure supplement 1.** Functional characterization of RPE1 stable clones.

phospho-Tyr availability or short ligand-receptor complex half-life, STAT3 activation is more robust than STAT1 activation.

To further support the STAT1/3 competition model, we next evaluated whether modulation of the STAT3/STAT1 ratio would influence STAT1 and STAT3 phosphorylation levels triggered by the different IL-6 ligands. To this end, we first studied the effect of decreasing the STAT3 levels on STAT1 activation. For that, HeLa cells were transfected with either scrambled siRNA or siRNA specific for STAT3 and then stimulated with the different IL-6 variants. STAT3 levels were decreased by more than 80% in transfected cells (*Figure 5—figure supplement 1c*). As expected, cell lacking STAT3 expression fail to induce STAT3 phosphorylation (*Figure 5e*). While STAT3 silencing did not change the STAT1 activation levels induced by HyIL-6 and Mut3, STAT1 activation was significantly increased upon C7 stimulation, suggesting a competition between STAT1 and STAT3 for Tyr in gp130 (*Figure 5e*). We then investigated whether increasing the levels of STAT1 protein would negatively affect STAT3 activation induced by IL-6. For this purpose, HeLa cells were transiently transfected with STAT1-GFP and the levels of STAT1 and STAT3 phosphorylation were assayed after 15 min of stimulation with 100 nM of the different IL-6 ligands by flow cytometry (*Figure 5f*). We took advantage of flow cytometry and the GFP-tag on STAT1 to gate HeLa cells expressing different amounts of STAT1 and studied STAT1 and STAT3 phosphorylation in each gated population (*Figure 5—figure supplement 1e*). As shown in *Figure 5f*, as the levels of STAT1 increased, so did the levels of phospho-STAT1 induced by HyIL-6 stimulation. This effect was not evident for the lower affinity ligands Mut-3, C7 and A1. The opposite results were observed when we analysed STAT3 activation levels. As the levels of STAT1 protein increased, the amount of phospho-STAT3 activated by the different IL-6 ligands decreased, confirming a competition between these two STAT molecules for phospho-Tyr on gp130. Overall, our data strongly support a model where ligand-receptor complex half-life and STATs competition for phospho-Tyr on cytokine receptor intracellular domains maintain a tight equilibrium that allow cells to fine tune their signaling output upon cytokine stimulation.

## IL-6 variants induce graded STAT3 transcriptional responses

Our data clearly indicate that cytokine-receptor complex half-life instructs biased signaling output by cytokine receptors. However, whether the observed changes in signaling ultimately translate into proportional gene expression changes and bioactivities is not clear. To investigate the immediate effects of 'biased' IL-6 signaling input on transcriptome of immune cells, we have generated global transcriptional profiles elicited by the three IL-6 variants in human Th1 cells. First, we performed signaling experiments in human Th1 T cells, to confirm signaling biased by the IL-6 variants in cells expressing gp130 and IL-6Rα receptor subunits simultaneously. Purified human CD4 T cells were activated through its T cell receptor (TCR) in vitro and expanded in Th1 polarizing conditions for five days before they were stimulated with saturating doses of HyIL-6 or the three IL-6 variants. As in HeLa cells, Mut3 activated STAT1 and STAT3 to the same extent as HyIL-6 in Th1 cells (*Figure 6—figure supplement 1a-b*). C7 and A1 activated STAT1 and STAT3 to different extents with C7 activating 60% STAT1 and 85% STAT3 when compared to HyIL-6, and A1 activating 50% STAT1 and 70%

STAT3 when compared to HyIL-6 (*Figure 6—figure supplement 1a-b*). These resulted in an increased STAT3/STAT1 activation ratio by C7 and A1 variants (*Figure 6—figure supplement 1c*).

Accordingly, to quantify its effects on gene expression, Th1 cells were stimulated with saturated concentrations of the three IL-6 variants and HyIL-6 for six hours to ensure that the entire cell population respond uniformly to the respective cytokine stimulation and their gene expression program analyzed by RNA-seq studies. We detected the upregulation of 23 genes in response to all four IL-6 variants (Fold change >1.5, FDR < 0.05, RPKM >4; *Figure 6a–b*), which were all classical STAT3-induced genes, validating the RNA-seq study. Importantly, most target genes showed a graded increase in the rate of transcription as a function of increasing STAT3 activity as exhibited by the three IL-6 variants (pSTAT3 levels; hyIL-6 = Mut3>C7>A1). However, the magnitude of transcriptional outputs differs widely from gene to gene, with some genes achieving maximal transcript levels even at low STAT3 levels (*Figure 6b*). For instance, while Mut3 gene signature resembles that of HyIL-6, C7 and A1 variants gene signatures exhibited a graded response, with some gene induction decreased by 50% (e.g. *SOCS3* and *BCL3*) and other genes expression barely affected when compared to HyIL-6 (e.g. *JAK3*, *ANK3*, *PIM2*). We further confirmed these observations by qPCR studies (*Figure 6—figure supplement 1d-g*). Overall these results suggest that IL-6-induced genes are differently sensitive to corresponding changes in nuclear STAT3 levels, which could provide the cell with the necessary flexibility to fine-tune its responses to wide-range of cytokines levels.

Next, to investigate how IL-6-induced STAT3 sites within the genome orchestrate the observed graded gene expression response, we measured global STAT3 binding profiles by ChIP-seq and compared the transcriptional activity of its target genes. Specifically, given that IL-6 variants induced different levels of STAT3 phosphorylation, we quantified genome-wide STAT3 binding sites in Th1 cells as a function of gradient STAT3 activation by the IL-6 variants. As expected, IL-6 stimulation led to STAT3 binding to 3480 genomic loci (*Figure 6d*), which were localized near classical STAT-associated genes (*Figure 6e*). We could detect significant changes in STAT3 binding intensity in response to the different IL-6 variants, which correlated with their STAT3 activation levels (*Figure 6f*). Of note, although ChIP-seq data identified many genome-wide IL-6-induced STAT3 binding sites, only a handful of those STAT3-target genes (23 transcripts) were upregulated in Th1 cells, suggesting additional mechanisms by which IL-6-induced STAT3 influences gene expression programs. Moreover, when we examined STAT3 bound regions near genes upregulated by IL-6 stimulation (*Figure 6c*), we observed a similar trend to that observed in the RNA-seq studies, that is STAT3 binding intensities were more different in those genes differentially regulated by the IL-6 variants (eg. *BCL3* and *SOCS3*), and more similar in genes equally regulated by all four ligands (e.g. *JAK3* and *PIM2*) (*Figure 6g–i* and *Figure 6—figure supplement 1h-i*). Interestingly, *SOCS3* and *BCL3* that were among the most differentially expressed IL-6-induced genes, contain multiple STAT3 binding sites (Supplemental Table 1), which may enable IL-6 to produce graded transcriptional outputs among its target genes. By contrast, STAT3 target genes with 1 or two binding sites at the gene promoter become saturated at relatively low levels of STAT3 transcriptional activation. This suggests that genes with multiple STAT3 binding sites would be more sensitive to changes in STAT3 signaling levels compared to gene with a single STAT3 binding site. Collectively, our data indicates that IL-6 variants result in graded STAT3 binding and transcriptional responses.

## IL-6 variants induce immuno-modulatory activities with different efficiencies

IL-6 is a highly immuno-modulatory cytokine, contributing to the inflammatory response by inducing differentiation of Th17 cells and inhibition of Treg and Th1 cells (*Heink et al., 2017*; *Jones et al., 2010*; *Kimura and Kishimoto, 2010*; *Louten et al., 2009*) (*Figure 7a–c*). We next asked whether these three activities would be uniformly affected by the biased signaling programs engaged by the three IL-6 variants. For that, we cultured resting human CD4 T cells in Th17, Th1 and Treg polarizing conditions in the presence/absence of the different IL-6 variants. As shown in *Figure 7*, the three variants induced responses that parallel their STAT activation potencies (*Figure 7d–f*). However, not all three activities were equally engaged by the three IL-6 variants. While all variants induced differentiation of Th17 cells to some extent (*Figure 7d* and *Figure 7—figure supplement 1*), C7 and A1 variants struggle to inhibit differentiation of Treg and Th1 cells, with C7 eliciting some inhibition and A1 failing in both cases (*Figure 7e–f* and *Figure 7—figure supplement 1a-b*). This is better represented in *Figure 7g*, where a triangular illustration is used to show that Mut3 is equally potent in

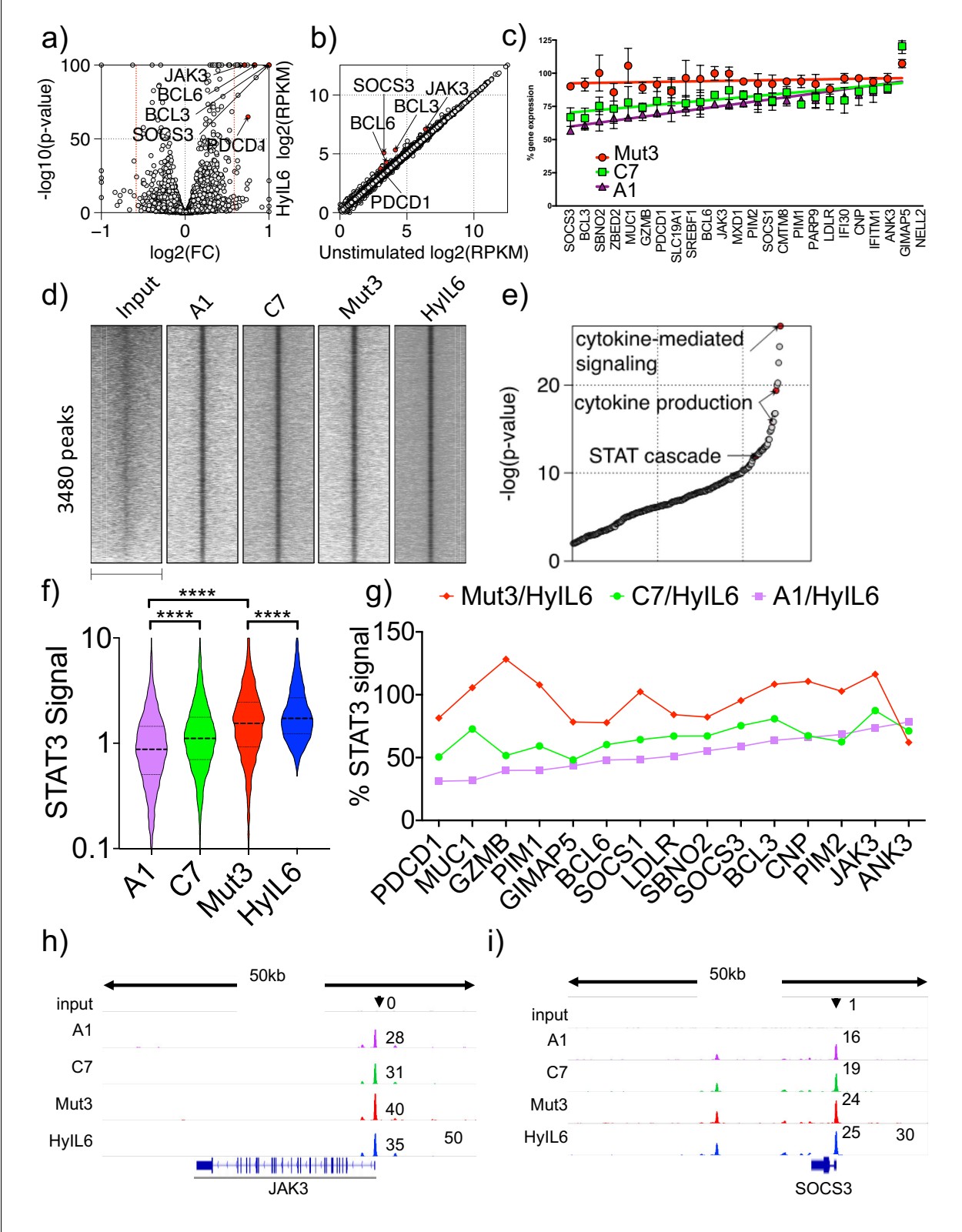

**Figure 6.** Transcriptional program elicited by the different IL-6 variants. (a) volcano plot showing significant genes differently expressed in Th1 cells after 6 hr stimulation with HyIL6. The red dash lines demark fold change = 1.5. (b) scatter plot showing mean gene expression values (n = 3) before (X-axis) and after HyIL6 stimulation (Y-axis). Top five differently expressed genes are highlighted. (c) plot showing the normalized gene expression relative to HyIL6 stimulation for each indicated stimulation. 23 differently expressed genes after HyIL6 stimulation are shown. The regression lines are

*Figure 6 continued on next page*

*Figure 6 continued*

highlighted. The data in **a–c**) are from three independent donors. (**d**) heatmap showing signal intensity of STAT3 bound regions (5 kb centred at peak summit) for indicated stimulations. Peaks are identified by comparing HyIL-6 stimulation and input. (**e**) shown are GO biological pathways ranked by p-value that are enriched in genes with adjacent STAT3 binding. (**f**) violin plot showing the signal intensity of all peaks (200 bp regions centred at peak summit) after each stimulation. P values are determined by two-tailed Wilcoxon tes (****p<0.0001). (**g**) shown are relative signal intensity of STAT3 peaks near select genes. Select are 15 differently expressed genes with adjacent STAT3 binding sites. (**h–i**) STAT3 binding at JAK3 (**h**) and SOCS3 (**i**) gene loci.

The online version of this article includes the following figure supplement(s) for figure 6:

**Figure supplement 1.** Transcriptional characterization of IL-6 variants.

inducing the three activities, producing an equilateral triangular shape. C7 and A1 on the other hand produced non-equilateral triangular shapes, exhibiting different induction efficiencies of the three bioactivities. Overall, these results show that not all cytokine bioactivities require the same signaling threshold, and that by modulating cytokine-receptor dwell times, we can decouple or at least bias cytokine-induced signaling programs to mediate specific cellular responses.

## Discussion

In this study we have engineered IL-6/gp130 binding kinetics to modulate signaling output and decouple IL-6 functional pleiotropy. Two main findings arise from our study: (1) Intracellular traffic dynamics of cytokine-receptor complexes and STATs binding affinities for phospho-Tyr on cytokine receptor ICD act synergistically to define signaling potency and identity, and (2) cells exhibit different gene induction thresholds in response to cytokine partial agonism, which allow them to modulate their responses. The current work, together with previous studies describing signaling tuning in other cytokines systems (*Ho et al., 2017*; *Kim et al., 2017*; *Moraga et al., 2015b*; *Mitra et al., 2015*), outline a general strategy to design cytokine partial agonists and decouple cytokine functional pleiotropy by modulating cytokine-receptor binding kinetics.

All our IL-6 variants must dimerize gp130 to some extent, since they all trigger signaling. However, our single molecule TIRF data show that low affinity variants struggle to promote detectable gp130 dimerization. These data suggest that non-detectable short-lived IL-6/gp130 complexes can partially engage signaling, but fail to trigger a full response, evoking a kinetic proof reading model. A kinetic proof reading model has been previously proposed for other ligand-receptor systems, including the T cell receptor system (TCR) (*McKeithan, 1995*) and more recently Receptor Tyrosine Kinases (RTKs) (*Zinkle and Mohammadi, 2018*). In these two systems, changes on ligand-receptor complex dwell-times induce phosphorylation of different Tyr pools in the receptors ICDs, ultimately recruiting and activating different signaling effectors (*Acuto et al., 2008*; *Lemmon and Schlessinger, 2010*). More recently, this model was used to explain biased signaling triggered by an EPO mutant (*Kim et al., 2017*). However, cytokine receptors differ significantly from RTKs and the TCR. While in these latter receptor systems, activation of different signaling effectors is clearly assigned to phosphorylation of specific Tyr in their ICDs, this is not generally true for cytokine receptors. Often only one or two Tyr in the cytokine receptor ICDs are required for signal activation (*Cheng et al., 2011*; *Schmitz et al., 2000*; *Zhao et al., 2008*).

How can short-lived cytokine-receptor complexes engage different signaling effectors that compete for a single phospho-Tyr? Our study provides new molecular evidences suggesting that STAT activation by cytokine-receptor complexes is governed by a kinetic discrimination mechanism. STATs compete for Tyr in receptors ICDs, thus making them sensitive to changes in cytokine-receptor complex dwell-times. STATs binding with low affinity to phospho-Tyr require longer-lived cytokine-receptor complexes and higher ligand doses to reach maximal activation. In agreement with this model, a previous study reported STAT1 and STAT3 binding with different affinities to phospho-Tyr in gp130 ICD (*Wiederkehr-Adam et al., 2003*). Moreover, using chimeric receptors and siRNA and overexpression approaches, we show that binding affinity of STAT proteins for phospho-Tyr in receptors ICDs defines signaling amplitude and identity by IL-6. Importantly, this is not the first evidence of STATs competing for phospho-Tyr. IFNα2 activates all STATs molecules, which can be abrogated by a single Tyr mutation in the IFNAR2 ICD, suggesting STAT competition (*Zhao et al., 2008*). Furthermore, modulation of STAT protein levels has been described to change signaling specificity by

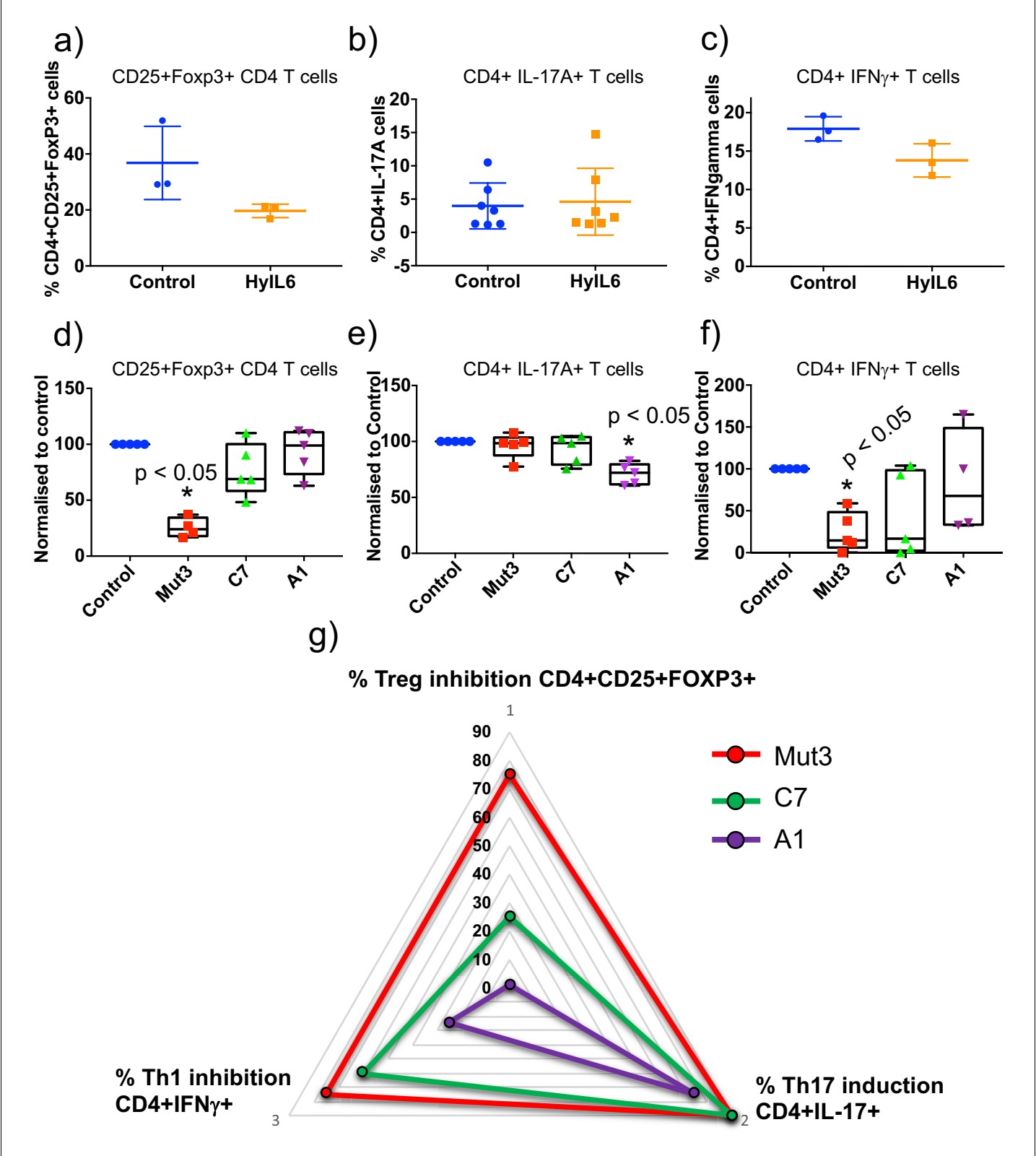

**Figure 7.** Immuno-modulatory activities trigger by the different IL-6 variants. (a) Human CD4 T cells were isolated from whole PBMCs and treated with Treg polarizing conditions in the presence of saturating concentrations of the different IL-6 variants for five days. Percentage of Treg cells were calculated by counting number of events in the CD4+CD25+FoxP3+ population obtained by flow cytometry. The control condition was defined as 100% response and the other conditions normalized accordingly. Data are mean + / - SEM from five independent replicates. (b) Human CD4 T cells

*Figure 7 continued on next page*

*Figure 7 continued*

were isolated from whole PBMCs and treated with Th17 polarizing conditions in the presence of saturating concentrations of the different IL-6 variants for fourteen days. Percentage of Th17 cells were calculated by counting number of events in the CD4+IL-17A+ population obtained by flow cytometry. The control condition was defined as 100% response and the other conditions normalized accordingly. Data are mean +/- SEM from five independent replicates. (c) Human CD4 T cells were isolated from whole PBMCs and treated with Th1 polarizing conditions in the presence of saturating concentrations of the different IL-6 variants for five days. Percentage of Th1 cells were calculated by counting number of events in the CD4+IFNγ+ population obtained by flow cytometry. The control condition was defined as 100% response and the other conditions normalized accordingly. Data are mean +/- SEM from four independent replicates. (d) Triangular representation of data from (a–c). As the affinity for gp130 decreases (C7 and A1 variants) the different IL-6 activities are differentially affected with Th17 differentiation being the most robust activity to changes in affinity and Treg inhibition being the most sensitive activity.

The online version of this article includes the following figure supplement(s) for figure 7:

**Figure supplement 1.** Immuno-modulatory properties of the IL-6 variants.

cytokines. IFNγ priming, which result in enhanced STAT1 protein levels, shift the IL-10 response from STAT3 activation to STAT1 activation (*Herrero et al., 2003*). In cells lacking STAT3, IL-6 switches to STAT1 activation, producing IFNγ-like responses (*Costa-Pereira et al., 2002*).

An alternative model that could explain our observations is that our IL-6 variants exhibit altered gp130 binding topology. Previous studies have shown that changes on cytokine-receptor binding topology, achieved using cytokine surrogate ligands, resulted in biased signaling programs (*Livnah et al., 1998*; *Moraga et al., 2015b*). We cannot formally exclude this possibility for our engineered IL-6 variants. The mutations introduced on the IL-6 variants to enhance their affinity for gp130 could have impacted their receptor binding topologies and therefore their signaling properties. However, we believe this possibility is unlikely. First, the mutations engineered in our variants are located exclusively on the IL-6-gp130 site-2 binding interface, leaving the cytokine backbone untouched and therefore greatly reducing the possibility of alterations on ligand-receptor binding topology. Second, there is a large body of literature showing that mutations in cytokine-receptor binding interfaces do not alter receptor binding topology (*Ho et al., 2017*; *Kim et al., 2017*; *Mendoza et al., 2017*; *Mitra et al., 2015*; *Moraga et al., 2015a*). Indeed, to our knowledge, there is not a single example in the literature of a cytokine mutant binding with alternative topology to its receptor. Third, our TIRF microscopy experiments confirm a canonical stoichiometry of the complex formed by the different IL-6 variants, excluding the formation of higher order oligomers that could result from alternative binding topologies (*Figure 3c*, and *Figure 3—figure supplement 1d*). Overall, our data support a kinetic proof reading model for cytokine signaling, where cytokine-receptor dwell times determine signaling output by kinetically discriminating activation of STAT molecules based on their binding affinity for phospho-Tyr on cytokine receptors ICDs. One key requirement for kinetic proof reading is the existence of an out-of-equilibrium process and rate-limiting kinetic intermediates (*Huang et al., 2016*; *Huang et al., 2019*). Importantly, the IL-6 system also follows this general principle. Internalization of the IL-6-gp130 signaling complex acts as a critical event that breaks the equilibrium by irreversibly capturing intact signaling complexes into endosomes. We have recently quantitatively described this effect in a systematic study with engineered IL-13 variants (*Moraga et al., 2015a*), showing the critical role of interaction rate constants in such a non-equilibrium process. For the engineered gp130 agonists, we clearly observed that endocytosis is altered, which was presumably caused by differential stability of ligand-receptor interaction (i.e. the off-rate), further supporting kinetic proof reading as the main mechanism for cytokine signaling diversification.

Initially thought to contribute to cytokine signaling shutdown, the endosomal compartment has emerged in recent years as a signaling hub, not only in cytokines (*Becker et al., 2010*; *Bulut et al., 2011*; *Claudinon et al., 2007*; *German et al., 2011*; *Keeler et al., 2007*; *Shah et al., 2006*) but also in other ligand-receptor systems (*Villaseñor et al., 2016*). Previous studies showed that activated JAK and STATs molecules are found in endosomes upon cytokine stimulation (*Payelle-Brogard and Pellegrini, 2010*), suggesting that cytokine-receptor complexes exhibit a signaling continuum from the cell surface to intracellular compartments. In agreement with this model, recent studies showed that cytokine-receptor complexes traffic to the endosomal compartment, where they are stabilized, contributing to signaling fitness (*Gandhi et al., 2014*; *Moraga et al., 2015a*). Only short-lived complexes that fail to traffic to intracellular compartments trigger diminished signaling output. Our data

fully support this model and expand it by showing that short-lived complexes that fail to traffic to the endosomal compartment engage a biased signaling program. Low affinity IL-6 variants, which did not induce internalization of gp130, activated more efficiently STAT3 than STAT1. It is possible that the stabilization of short-lived IL-6-gp130 complexes in endosomes provides them with the necessary extra time to activate secondary pathways that engage the receptor with low affinity. An alternative possibility would be that the high density of cytokine receptor phospho-Tyr motifs found in endosomes, as a consequence of the accumulation of cytokine-receptor complexes in the confined endosomal space (*Moraga et al., 2015a*), could diminish the advantage exhibited by signaling intermediates binding with high affinity cytokine receptors, therefore favouring the activation of secondary low affinity signaling molecules. Further studies will be required to understand kinetics of signal activation by cytokines in intracellular compartments. An open question pertains to whether the intracellular localization of STATs influence signaling by long- or short-lived cytokine receptor complexes. Early studies described the localization of STAT3 in intracellular membranes, while other STATs association with intracellular membranes is not so well described (*Shah et al., 2006*). Whether signaling by cytokines can be further engineered by modulation of the intracellular localization of STATs requires further investigation.

In the current study, we show that different genes downstream of IL-6 signaling exhibit different thresholds of activation that can be exploited by IL-6 partial agonists to decouple IL-6 immuno-modulatory activities. How these activation thresholds are established is not clear. STAT proteins face two points where the law of mass action influences their responses the most. The first one pertains to the binding of STATs to phospho-Tyr in the receptors ICDs, and as discussed above, contributes to define signaling potency and identity by cytokine-receptor complexes. The second point is found when activated STATs bind specific Gamma interferon Activated Sequences (GAS) motifs in the promoters of responsive genes. GAS sequences, although conserved, exhibit degrees of degeneracy that allow them to bind STATs molecules with different affinities (*Bonham et al., 2013*; *Ehret et al., 2001*; *Horvath et al., 1995*). In addition, different number of GAS sequences are found in different responsive promoters. In principle, the combination of STAT binding affinities for GAS sequences and the number of GAS sequence present in the promoters could generate different gene induction thresholds. In agreement with this model, we identified chromatin regions through our STAT3 Chip-Seq studies, that bound STAT3 with different efficiencies, with regions where STAT3 binding was diminished by changes in STAT3 activation levels, and regions where efficient STAT3 binding was detected in all conditions tested. When we analysed the number of GAS sequences and their motifs under those regions, we could detect that genes that were more sensitive to changes in STAT3 phosphorylation presented higher number of GAS motifs than those more resistant. Overall our data support a kinetic-proof reading model for cytokine signaling, whereby cytokine-receptor dwell time and STAT binding affinities for phospho-Tyr on receptors ICDs define potency and identity of cytokine signaling signatures. As the number of STAT molecules activated by partial agonists increases, additional GAS binding motifs are engaged in promoters with multiple GAS binding sites, triggering the induction of graded gene expression responses. In principle, engineering of cytokine-receptor binding kinetics could rescue cytokine-based therapies, by decreasing cytokine functional pleiotropy and toxicity.

## Materials and methods

### Protein expression and purification

Human IL-6 wild type and IL-6 variants were cloned into the pAcGP67-A vector (BD Biosciences) in frame with an N-terminal gp67 signal sequence and a C-terminal hexahistidine tag, and produced using the baculovirus expression system, as described in *LaPorte et al. (2008)*. Baculovirus stocks were prepared by transfection and amplification in *Spodoptera frugiperda* (*Sf*9) cells grown in SF900II media (Invitrogen) and protein expression was carried out in suspension *Trichoplusiani ni* (High Five) cells grown in InsectXpress media (Lonza). Following expression, proteins were captured from High Five supernatants after 48 hr by nickel-NTA agarose (Qiagen) affinity chromatography, concentrated, and purified by size exclusion chromatography on a Enrich SEC 650 1 × 300 column (Biorad), equilibrated in 10 mM HEPES (pH 7.2) containing 150 mM NaCl. Recombinant cytokines were purified to greater than 98% homogeneity. For biotinylated gp130 expression, gp130

ectodomain (SD1-SD3, amino acids 23–321) was cloned into the pAcGP67-A vector with a C-terminal biotin acceptor peptide (BAP)-LNDIFEAQKIEWHW followed by a hexahistidine tag. Purified Gp130 was in vitro biotinylayed with BirA ligase in the presence of excess biotin (100 µM). HyIL-6 was site-specifically labeled via an ybbR-tag by enzymatic phosphopantetheinyl-transfer using coenzyme A conjugates as described previously (*Waichman et al., 2010*). For site-specific fluorescence labeling of IL-6 variants with different fluorochromes, an accessible cysteine was introduced at the C-terminal of the cytokine and cloned in the pAcGP67-A vector as described above. Labeling of purified proteins was carried out with excess DY 647 and DY 547 maleimide, repectively in the presence of 50 µM TCEP. For single molecule labeling small, antigen-binding, single domain polypeptides which are derived from the variable heavy chain of the heavy-chain only antibodies of camelids were used (Nanobody, short NB) (*Hamers-Casterman et al., 1993*). NB was cloned into pET-21a with an additional cysteine at the C-terminus for site-specific fluorophore conjugation. Furthermore, a (PAS)$_5$ sequence to increase protein stability and a His-tag for purification were fused at the C-terminus. Protein expression in *E. coli* Rosetta (DE3) and purification by immobilized metal ion affinity chromatography was carried out by standard protocols. Purified protein was dialyzed against HEPES pH 7.5 and reacted with a two-fold molar excess of DY-647P1 (DY647) maleimide (Dyomics), and ATTO Rho11 (Rho11) maleimide (ATTO-TEC GmbH), respectively. After 1 hr, a 3-fold molar excess (with respect to the maleimide) of cysteine was added to quench excess dye. Protein aggregates and free dye were subsequently removed by size exclusion chromatography (SEC). A labeling degree of 0.9-1:1 fluorophore:protein was achieved as determined by UV/Vis spectrophotometry.

## Plasmid constructs

For single molecule fluorescence microscopy, monomeric non-fluorescent (Y67F) variant of eGFP (''mXFP'') was N-terminally fused to gp130. This construct was inserted into a modified version of pSems-26 m (Covalys) using a signal peptide of Igk. The ORF was linked to a neomycin resistance cassette via an IRES site. A mXFP-IL-27Rα construct was designed likewise. The chimeric construct mXFP-IL-27-Rα (ECD)-gp130(ICD) was a fusion construct of IL-27Rα (aa 33–540) and gp130 (aa 645–918). For mXFP-IL-27Rα(ECD)-gp130(ICD) ΔY the ICD of gp130 was truncated downstream of the JAK1 binding motif (aa 645–705).

## Cell lines and media

HeLa cells were grown in DMEM containing 10% v/v FBS, penicillin-streptomycin, and L-glutamine (2 mM). RPE1 cells were grown in DMEM/F12 containing 10% v/v FBS, penicillin-streptomycin, and L-glutamine (2 mM). HepG2 cells and Ba/F3-gp130 (*Gearing et al., 1994*) cells were cultured in DMEM containing 10% v/v FBS, and penicillin-streptomycin. Viability of Ba/F3-gp130 cells was determined as described previously (*Garbers et al., 2011*). Human T-cells were cultivated in RPMI supplemented with 10% v/v FBS, penicillin-streptomycin and cytokines for proliferation/differentiation (see below). RPE1 cells were stably transfected by mXFP- IL-27-Rα and the chimeric constructs by PEI method according to standard protocols. Using G418 selection (0.6 mg/ml) individual clones were selected, proliferated and characterized. For comparing receptor cell surface expression levels, cells were detached using PBS+5 mM EDTA, spun down (300 g, 5 min) and incubated with αGFP-nanobody$^{Dy647}$ (10 nM, 15 min on ice). After incubation, cells were washed with PBS and run on cytometer.

## CD4+ T cell purification

Peripheral blood mononuclear cells (PBMCs) of healthy donors were isolated from buffy coat samples (Scottish Blood Transfusion Service) by density gradient centrifugation according to manufacturer's protocols (Lymphoprep, STEMCELL Technologies). From each donor, $100 \times 10^6$ PBMCs were used for isolation of CD4+ T-cells. Cells were decorated with anti-CD4+$^{FITC}$ antibodies (Biolegend, #357406) and isolated by magnetic separation according to manufacturer's protocols (MACS Miltenyi) to a purity >98% CD4+.

## Flow cytometry staining and antibodies

For measuring dose-response curves of STAT1/3 phosphorylation (either TH1 cells or HeLa/RPE1 clones), 96-well plated were prepared with 50 µl of cell suspensions at $2 \times 10^6$ cells/ml/well for TH1 and $2 \times 10^5$ cells/ml/well for HeLa/RPE1. RPE1 cells were detached using Accutase (Sigma). Cells

were stimulated with a set of different concentrations to obtain dose-response curves. To this end cells were stimulated for 15 min at 37°C with the respective cytokines (mIL27sc or hypIL6) followed by PFA fixation (2%) for 15 min at RT.

For kinetic experiments, cell suspensions were stimulated with a defined, saturating concentration of cytokines (2 nM mIL27sc (murine single chain variant of EBI3 and p28), 10 nM hypIL6, 100 nM IL-6 mutants) in a reverse order so that all cell suspensions were PFA-fixed (2%) at the same time.

## Permeabilization, fluorescence barcoding and antibody staining

After fixation (15 min at RT), cells were spun down at 300 g for 6 min at 4°C. Cell pellets were resuspended and permeabilized in ice-cold methanol and kept for 30 min on ice. After permeabilization cells were fluorescently barcoded according to *Krutzik and Nolan (2006)*. In brief: using two NHS-dyes (PacificBlue, #10163, DyLight800, #46421, Thermo Scientific), individual wells were stained with a combination of different concentrations of these amino-reactive dyes. After barcoding, cells can be pooled and stained with anti-pSTAT1[Alexa647] (Cell Signaling Technologies, #8009) and anti-pSTAT3[Alexa488] (Biolegend, #651006) at a 1:100 dilution in PBS+0.5%BSA. T-cells were also stained with anti-CD8[AlexaFlour700] (Biolegend, #300920), anti-CD4[PE] (Biolegend, #357404), anti-CD3[BrilliantViolet510] (Biolegend, #300448). Cells were probed at the flow cytometer (Beckman Coulter, Cytoflex S). Individual cell populations were identified by their barcoding pattern and mean fluorescence intensity (MFI) of pSTAT1[647] and pSTAT3[488] was measured for all individual cell populations.

## Western blotting protocol

Cells were rinsed in ice-cold PBS then lyzed in NP40 lysis buffer (1% NP40, 50 mM Tris-HCl pH 8.0, 150 mM NaCl) plus protease inhibitor cocktail (Pierce), 5 mM sodium fluoride, 2 mM sodium orthovanadate and 0.2 mM PMSF incubating on ice for 15 min. Lysates were cleared by centrifugation at 20,000 g for 15 min at 4°C then protein concentrations determined using Coomassie Protein Assay Kit (Thermo Scientific, UK). For each sample, 30 μg of total protein were separated on 7% Bis-Tris polyacrylamide gels in SDS running buffer then blotted onto Protran 0.2 mM Nitrocellulose (GE Healthcare, UK). Membranes were probed with 1:1000 dilution of the appropriate primary antibody (mouse anti-gp130; Santa Cruz sc376280), rabbit anti-Clathrin (Biolegend, 813901), STAT3 (Cell Signaling Technologies, #9139), P-STAT3 (Y705, Biolegend, #651006) P-STAT1 (Y701, Cell Signaling Technologies, #8009) or 1:5000 dilution mouse anti-GAPDH (Cell Signaling Technologies, #2118), then 1:5000 dilution of donkey anti-rabbit-HRP (Stratech, 711-035-152-JIR) or donkey anti-mouse-HRP (Stratech, 715-035-150-JIR) as the secondary antibody. Immobilon Western Chemiluminescent HRP substrate (Millipore, UK) was used for visualization.

## siRNA silencing

HeLa cells were seeded at $2 \times 10^5$ cells per well in a six well plate and transfected with Clathrin siRNA (Oligo 1:AGGUGGCUUCUAAAUAUCAUGAACA; Oligo 2: GAAUGUUUACUGAAUUAGCUAUUCT sequences; from IDT Technology), STA3 siRNA (CAACAUGUCAUUUGCUGAA) or non-targeting siRNA (UGGUUUACAUGUCGACUAA) as a control (Dharmacon) using DharmaFect one transfection reagent (Dharmacon, Cat#T2001-02) following the manufacturer instructions. 48 hr later cells were treated as indicated and samples were prepared for immunoblotting analysis to check the level of gene knock-down and for Flow cytometry (FACS) (STAT3).

## IL6 variants internalization

HeLa gp130 KO cells were transiently transfected with a plasmid encoding for gp130-meGFP. 24 hr after transfection cells were stimulated for 30 min on ice with different concentrations of the ligands (dose-response experiments) or 25 nM HyIL6 and 100 nM of the IL6 variants (*i.e.* Mut3, C7, A1) (kinetics experiments).

For the dose-response experiments, upon 30 min incubation on ice, cells were incubated at 37°C for 30 min. Then, cells were washed a couple of times with PBS, incubated with trypsin for 10 min at 37°C to remove all the ligand bound to the cell surface but not internalized. Cells were then washed again, resuspended in DMEM and fixed with PFA 2% for 20 min. Finally, the levels of internalization of the different ligands were analyzed by flow cytometry (CytoFlex).

Regarding the kinetics experiments upon 30 min incubation on ice, cells were incubated at 37°C for the indicated times. Then, cells were washed a couple of times with PBS, incubated with trypsin for 10 min at 37°C to remove all the ligand bound to the cell surface but not internalized. Cells were then washed again, resuspended in DMEM and fixed with PFA 2% for 20 min. Finally, the levels of internalization of the different ligands were analyzed by flow cytometry (CytoFlex).

## IL6 variants-induced gp130 uptake and internalisation

HeLa WT cells ($2 \times 10^6$ per condition) were incubated with cycloheximide (5 µg/ml) for 30 min to block protein synthesis. Then, cells were stimulated with saturating concentrations of the different ligands (20 nM HyIL6 or 100 nM Mut3, C7 or A1) for the indicated times. Cells were then lysed in RIPA buffer containing protease inhibitors (ROCHE) and the total lysates were used for immublotting analysis (anti-gp130, Santa Cruz (sc-376280)) as described above.

## Effect of clathrin inhibition (Pitstop) on HyIL6-Dy647 internalisation

HeLa gp130 KO cells transiently transfected with a plasmid encoding for gp130-meGFP, were pre-incubated with 60 µM Pitstop (Sigma) for 30 min. Then, cells were incubated with 20 nM HyIL6-Dy647 on ice for 30 min, and finally incubated at 37°C for 0, 15, 30, 60 or 90 min in the presence of the ligand. Later, cells were treated with trypsin in order to remove the ligand not internalized, fixed with PFA 2% for 20 min at room temperature and analysed by flow cytometry (CytoFlex).

## Gp130 Tyr-Phe mutants signaling experiments

Gp130 mutants were designed to substitute the Tyr residues previously reported as required for the phosphorylation of STAT1 and STAT3 (namely Y757, Y815, Y905 and Y915). These gp130 mutants were single mutants-Y757F, Y815F, Y905F and Y915F-, double mutants in which the two more distal Tyr residues were replaced by Phe (2F-Y905F+Y915F), triple mutants (3F-Y815F-Y905F+Y915F) and finally a gp130 mutant in which all the Tyr residues were replaced by Phe (4F-Y757F-Y815F-Y905F +Y915F). HeLa gp130 KO cells were transfected ON with these different constructs, and then used to analysed the requirement of the different Tyr residues to drive the phosphorylation of both STAT1 and STAT3 upon stimulation with different concentrations of HyIL6 for 15 min. Then, cells were fixed with PFA 2%, permeabilized with methanol 100% on ice for 30 min, stained for P-STAT1 (CellSignaling, 8009S) or P-STAT3 (BioLegend, 651007) and analyzed by flow cytometry (CytoFlex).

## HeLa gp130 crispr/CAS9 knock-out generation

5 µl of 200µM Alt-R CRISPR -Cas9 crRNA (IDT, Hs.Cas9.IL6ST.1.AF) and 5 µl of 200µM Alt-R CRISPR-Cas9 tracrRNA (IDT) were combined, heated to 95°C for 5 min then the tube cooled to room temperature. 1.2 µl RNA duplex was mixed with 1.7 µl Alt-R S.p. HiFi Cas9 Nuclease V3 (IDT) and 2.1 µl PBS and these incubated at RT for 20 min. $4 \times 10^5$ HeLa cells resuspended in 8 µl of buffer R (Neon Transfection System Kit, Thermo) were added to the tubes with the RNP complexes. The electroporation parameters used were: two pulses of 1,005 V with a pulse width of 35 ms. Then, reactions were added directly into antibiotic-free media in a well of the 6-well plate and incubated at 37°C for 16 hr. HeLa cells electroporated with RNP particles were transferred into DMEM media containing 10% FCS and Pen/Strep, expanded and finally individual clones were isolated and tested for gp130 knock-out.

## Assembly, transformation, and selection of the IL-6 library

Yeast surface display protocol was adapted from previously described ones (*Boder and Wittrup, 1997*). Human IL-6 cDNA was cloned into the yeast display vector pCT302. *S. cerevisiae* strain EBY100 was transformed with the pCT302_IL-6 vector. Generally, yeast were grown in SDCAA media pH: 4.5 for one day, and induced in SGCAA media pH: 4.5 for two days, before undergoing a round of selection. Different concentrations of biotinylated gp130 ectodomains were used to carried out the selections. In initial rounds where gp130-Streptavidin (SA) tetramers were used to select low affinity gp130 binders, tetramers were formed by incubating gp130 and SA coupled to Alexa-647 dye at a ratio of 4:1 gp130:SA for 15 min on ice.

The assembly of the library DNA was carried out using 14 overlapping primers, two of which contained the NDT codon (G,V,L,I,C,S,R,H,D,N,F,Y) used for mutation. The following amino acids were

chosen to randomize: D9, E22, R23, K26, Q27, Y30, D33, G34, A37, E109, R112, M116, V120, F124. The PCR product was further amplified, to obtain 50 µg using the primers:

> 5'- TAGCGGTGGGGGCGGTTCTCTGGAAGTTCTGTTCCAGGGTCCGAGCGGCGGATCCGTACC CCCAGGAGAAGATTCC −3'
> 5'- CGAGCAAGTCTTCTTCGGAGATAAGCTTTTGTTCGCCACCAGAAGCGGCCGCCATTTGCCG AAGAGCCCTCAG −3'

These primers also contained the necessary homology to the pCT302 vector sequence requisite for homologous recombination. Insert DNA was combined with linearized vector backbone pCT302 and electrocompetent *S. cerevisiae* EBY100 were electroporated and rescued, as previously described, forming a library of $3 \times 10^8$ transformants. Selections were performed on this library using magnetic activated cell sorting (MACS, Miltenyi Biotech). The first round of selection was performed with $2 \times 10^9$ cells from the yeast library, approximately 10-fold coverage relative to the number of transformants. Subsequent rounds of selection used $1 \times 10^7$ yeast cells (greater than 10-fold coverage in each round). Fluorescence analysis was performed on a CytoFlex cytometer.

## Determination of binding kinetics by switchSENSE

All measurements were performed on a dual-color $DRX^2$ instrument using a standard switchSENSE chip (MPC2-48-2-G1R1, Dynamic Biosensors GmbH), which provides two differently labeled DNA sequences on each electrode (green fluorescent NL-A48, red fluorescent NL-B48). The chip was functionalized by initial hybridization of streptavidin-cNL-B48 conjugate and bare cNL-A48 DNA (each 200 nM, HE40 buffer, Dynamic Biosensors GmbH). In this way, the red fluorescence yields the signal for the interaction measurement with the target molecule, while the green fluorescence provides an on-spot reference for unspecific effects. In a second step, biotinylated gp130 was injected and captured onto the surface by immobilized streptavidin. To analyze the gp130 – IL-6 interactions, a series of protein concentrations (62 nM - 12 µM) was tested. All experiments were performed in HEPES-based running buffer (10 mM HEPES, 140 mM NaCl, 0.05% Tween20, 50 µM EDTA, 50 µM EGTA, pH = 7.4) at 25˚C. For measuring the association, IL-6 variants were injected with a flowrate of 500 µl/min between 60 and 120 s and the absolute fluorescence in static mode was recorded (fluorescence proximity sensing). Dissociations was monitored at the same flow rates (500 µl/min) and varied between 7 min and 3 hr depending on dissociation rate constants determined during assay development. After each cycle (analyte concentration), the surface was regenerated and freshly functionalized. Association and dissociation rates were determined by fitting a global mono-exponential model to the raw data.

## qPCR studies

Resting $CD4^+$ T cells were labeled with anti-CD4-FiTC antibody (BioLegend, Cat#357406) and isolated from human PBMCs by magnetic activated cell sorting (MACS, Miltenyi) using anti-FiTC microbeads (Miltenyi, Cat#130-048-701) following manufacturer instructions. Subsequently, resting $CD4^+$ T cells were activated under Th1 polarizing conditions. Briefly, $10^6$ resting human $CD4^+$ T cells per ml were primed for three days with ImmunoCult Human CD3/CD28 T Cell Activator (StemCell) following manufacturer instructions in the presence of IL2 (20 ng/ml, Novartis Cat#709421), IL12 (20 ng/ml, BioLegend, Cat#573002) and anti-IL4 (10 ng/ml, BD Biosciences, Cat#554481). Then, cells were expanded in the presence of IL2 (20 ng/ml) and anti-IL4 (10 ng/ml) for another 5 days. Cells were starved without IL2 for at least 24 hr before the stimulation with the different forms of IL6 for 6 hr. Total RNA was isolated using the RNeasy Mini Kit (Qiagen, Cat#74104) and equal amounts of cDNA were synthesised using the iScript cDNA Synthesis Kit (BioRad, Cat# 1708890). 100 ng of cDNA were used to assay the expression level of the different genes of interest by qPCR using TB Green Premix Ex Taq II (Takara, Cat# RR820L) in a CFX96 Touch Real-Time PCR Detection System (BioRad). GAPDH was amplified as an internal control. The relative quantitation of each mRNA was performed using the comparative Ct method and normalised to the internal control.

Primers for qPCR analysis were:

## GAPDH
Fw: 5'-ACCCACTCCTCCACCTTTGA-3' Rv: 5'-CTGTTGCTGTAGCCAAATTGGT-3',

SOCS3
Fw: 5'-GTCCCCCCAGAAGAGCCTATTA-3' Rv: 5'-TTGACGGTCTTCCGAGAGAGAT-3',

BCL3
Fw: 5'-GAAAACAACAGCCTTAGCATGGT-3' Rv: 5'-CTGCGGAGTACATTTGCG-3',

PIM2
Fw: 5'-GGCAGCCAGCATATGGG-3' Rv: 5'-TAATCCGCCGGTGCCTGG-3',

JAK3
Fw: 5'-GCCTGGAGTGGCATGAGAA-3' Rv: 5'-CCCCGGTAAATCTTGGTGAA-3'.

## Chromatin immunoprecipitation by sequencing (ChIP-Seq)

In vitro polarized human Th1 cells were expanded in the presence of IL-2 for 10 days and cells were then washed with complete media and rested for 24 hr starvation in the absence of IL-2, these cells were then either not-stimulated (control) or stimulated with IL-6 or different IL-6 variants for 1 hr, cells were then immediately fixed with 1% methanol-free formaldehyde (Formaldehyde 16%, Methanol-Free, Fisher Scientific, PA, USA) at room temperature for 10mn with gentle rocking cells were then washed twice with cold PBS. For each STAT3 ChIP-seq library sample, approximately $10 \times 10^6$ cells were used and the fixed cell palettes were kept at $-80°C$ prior to further processing. The ChIP-seq experiments were performed as previously described (*Liao et al., 2008*) with some modification as described below. In brief, the frozen cell pellets were thawed on ice and washed once with 1 mL cold PBS by centrifugation at 5000 RPM for 5 min, the resulting cell pellets were re-suspended in 500 uL of lysis buffer (1X PBS, 0,5% Triton X-100, cOmplete EDTA-free protease inhibitor cocktail, Roche Diagnostics, Basel, Switzerland) and incubated for 10 min on ice, followed by a 5 min centrifugation at 5000 RPM. Then the pellets were washed once with 1 mL of sonication buffer (1X TE, 1: 100 protease inhibitor cocktail), re-suspended in 750 uL of sonication buffer (1X TE, 1: 100 protease inhibitor cocktail and 0,5 mM PMSF) and sonicated for 20 cycles (on-20sec and off-45sec) on ice using VCX-750 Vibra Cell Ultra Sonic Processor (Sonics, USA). The sonicated lysates were centrifuged 20 min at 14000 RPM and the clear lysate supernatants were collected and incubated with 30 uL of Protein-A Dynabeads (ThermoFisher, USA) that were pre-incubated with incubated with 10 ug of anti-STAT3 antibody (anti-Stat3, 12640S, Cell Signaling Technology) at 4°C overnight with gentle rotation. Next day, the beads were washed 2 times with RIPA-140 buffer (0.1% SDS, 1% Triton X-100, 1 mM EDTA, 10 mM Tris pH 8.0, 300 mM NaCl, 0.1% NaDOC), 2 times with RIPA-300 buffer (0.1% SDS, 1% Triton X-100, 1 mM EDTA, 10 mM Tris, 300 mM NaCl, 0.1% NaDOC), 2 times with LiCl buffer (0.25 mM LiCl, 0.5% NP-40, 1 mM EDTA, 10 mM Tris pH 8.0, 0.5% NaDOC), once with TE-0,2% Triton X-100 and once with TE buffer. Crosslinks were reversed by incubating the bound complexes in 60 uL TE containing 4.5 uL of 10% SDS and 7.5 uL of 20 mg/mL of proteinase K (Thermofisher, USA) at 65°C overnight for input samples, we used 6 uL of 10% SDS and 10 μL of 20 mg/mL of proteinase K. Then, the supernatants were collected using a magnet and beads were further washed one in TE 0.5M NaCl buffer. Both supernatants were combined, and DNA was extracted with phenol/chloroform, followed by precipitation with ethanol and re-suspended in TE buffer. The library was constructed following the manufacturer protocol of the KAPA LTP Library Preparation Kit (KAPA Biosystems, Roche, Switzerland). ChIP DNA libraries were ligated with the Bioo scientific barcoded adaptors (BIOO Scientific, Perkin Elmer, USA) with T4 DNA ligase according to KAPA LTP library preparation protocol and the ligated ChIP DNA libraries were purified with 1.8x vol. Agencourt AMPure XP beads and PCR amplified using KAPA hot start High-Fidelity 2X PCR Master Mix and NextFlex index primers (Bioo Scientific, PerkinElmer) for 12 cycle by following thermocycler cycles: 30 s hot start at at 98°C, followed by 12 cycle amplification [98°C for 10 s, 60°C for 30 s and 72°C for 30 s] and final extension at 72°C for 1 min. The amplification and quality of the ChIPseq libraries were checked by running 10% of the samples in E-Gel Agarose Gels with SYBR Safe DNA Gel Stain (ThermoFisher Scientific, USA), and if necessary, samples were reamplified additional four cycles using the same thermocycler protocol described above. Then, the libraries were purified and size-selected using Agencourt AMPure XP beads (1.25x vol. to remove short fragments. The concentration of ChIP-DNA libraries was measured by Qubit-4 fluorometer (ThermoFisher, USA) and equal

amounts of each sample were pooled and 50 bp paired-end reads were sequenced on an Illumina 4000 platform by GENEWIZ technology (GENEWIZ, USA).

## RNA-sequencing

For RNA-seq library preparation, in vitro polarized human Th1 cells either not stimulated or stimulated with the different IL-6 variants at 37°C for 6 hr, total RNA was extracted and RNAseq libraries were prepared by Edinburg Sequencing Core facility.

## ChIP-seq data analysis

The quality of generated libraries was inspected using FastQC v0.11.8. All sequencing reads were aligned to human reference genome (GRCh37; hg19) using bowtie.v1.2.2[1] with default parameters except '-chunkmbs 1000 S -m 1'. The genome index was generated using 'bowtie-build' using default parameters. The aligned reads were indexed using samtools v1.9[2] for further processing. The genome-wide binding profile (i.e. Bigwig files) were generated by bamCoverage v3.2.0[3] using default parameters except '-normalizeUsing BPM -minMappingQuality 30 -ignoreDuplicates -extendReads 250 -blackListFileName hg19.blacklist.bed'. The binding profiles were visualized using IGV genome browser v2.5.0[4]. Binding peaks were called by 'callpeaks' procedure from MACS2 v2.1.2[5] using default parameters except '-f BAMPE -nomodel -t mutant -c input'. The identified peaks were further screened against 'hg19 blacklisted' genomic regions[6], mitochondrial DNA, and pseudo-chromosomes. The binding heatmap surrounding HyIL-6 bound regions was generated by ChAsE v1.0.11[7]. HyIL-6 bound regions were sorted by significance and annotated by 'annotatePeaks' procedure from HOMER v4.10[8] to obtain the nearest genes. Pathway analysis of the top 2000 annotated genes was performed by Metascape[9] on all GO terms related to biological processes. The resulting pathways were sorted by significance and plotted by Datagraph v4.3. The average binding signal intensity for each peak was calculated by UCSC bigWigAverageOverBed v2 using default parameters. De novo Motif findings were performed in 200 bp bound regions (n = 500) using MEME Suite v5.0.2[10] with default parameters except '-maxsize 10000000 -dna -mod zoops -nmotifs 10'. De novo motifs were compared against all JASPAR known motifs by TOMTOM[11]. Statistical analyses were performed using the indicated Two-tailed parametric and non-parametric tests as appropriate.

## RNA-seq data analysis

The quality of generated libraries was inspected using FastQC v0.11.8. The RNA expression level in each library was estimated by 'rsem-calculate-expression' procedure in RSEM v1.3.1[12] using default parameters except '-bowtie-n 1 -bowtie-m 100 -seed-length 28 -paired-end'. The bowtie index required by RSEM software was generated by 'rsem-prepare-reference' on all RefSeq genes, obtained from UCSC table browser on April 2017. EdgeR v3.24.0[13] package was used to normalize gene expression among all libraries and identify differentially expressed genes among samples with following constraints: fold change $\geq$1.5, FDR $\leq$ 0.05 and RPKM > 4 in at least one of two compared samples. The volcano plot representation was used to depict the log fold change of gene expression (HyIL-6 vs. unstimulated; n = 3) as a function of significance. The scatter plot was used to show the expression of genes in HyIL-6 stimulated against unstimulated samples. The expression values were the average of (n = 3) independents donors. Differentially expressed genes under HyIL-6 stimulations were probed for response by the three indicated mutants (i.e. Mut3, C7, A1). The expression values were normalized to HyIL-6 and plotted by PRISM v8.1.0.

## T cells population differentiation

Resting CD4$^+$ T cells isolated as described above were activated under Th1, Th17 or Tregs polarizing conditions. Briefly, resting human CD4$^+$ T cells freshly isolated were activated using ImmunoCult Human CD3/CD28 T Cell Activator (StemCell, Cat#10971) following manufacturer instructions for 3 days in the presence of the cytokines required for the different CD4$^+$ T cells populations: Th1 (IL-2 (20 ng/ml), anti-IL-4 (10 ng/ml), IL-12 (20 ng/ml)), Th17 (IL-1β (10 ng/ml, R and D Systems, Cat#201-LB/CF), IL-23 (10 ng/ml, R and D Systems, Cat#1290-IL), anti-IL-4 (10 ng/ml), anti-IFNγ (10 ng/ml, BD Biosciences, Cat#554698)) or Tregs (IL2 (20 ng/ml), TGF-β (5 ng/ml, Peprotech, Cat#100–21), anti-IL-4 (10 ng/ml), anti-IFNγ (10 ng/ml)) in the presence or absence of saturating concentrations of the

different variants of IL6 described in this manuscript. After three days of priming, cells were expanded for another 5 days in the presence of IL-2 (20 ng/ml). Th1 and Th17 cells were restimulated for 6 hr in the presence of PMA (100 ng/ml, Sigma, Cat#P8139), Ionomycin (1 µM, Sigma, I0634) and Brefeldin A (5 µg/ml, Sigma, B7651) before FACS analysis. In all cases cells were fixed with 2% formaldehyde and prepared to be analysed by FACS. Cells were then permeabilised with Saponin 2% in PBS for 20 min at room temperature and then stained in Saponin 2% in PBS with the appropriate antibodies: Th1 (anti-CD3-BV510 (1:100, Biolegend, Cat#300448), anti-CD4-PE (1:100, Biolegend, Cat#357404), anti-CD8-AF700 (1:100, Biolegend, Cat#300920), anti-IFNγ (1:100, Biolegend, Cat#502217)), Th17 (anti-CD3-BV510, anti-CD4-PE, anti-CD8-AF700, anti-IL17A-APC (1:100, Biolegend, Cat#512334)) and Tregs (anti-CD3-BV510, anti-CD4-PE, anti-CD8-AF700, anti-CD25-APC (1:100, Biolegend, Cat#302610), anti-FoxP3-AF488 (1:100, Biolegend, 320012)) and analysed in a CytoFLEX S (Beckman Coulter).

## Live-cell dual-color single-molecule imaging studies

Single molecule imaging experiments were carried out by total internal reflection fluorescence (TIRF) microscopy with an inverted microscope (Olympus IX71) equipped with a triple-line total internal reflection (TIR) illumination condenser (Olympus) and a back-illuminated electron multiplied (EM) CCD camera (iXon DU897D, 512 × 512 pixel, Andor Technology). A 150 x magnification objective with a numerical aperture of 1.45 (UAPO 150 3/1.45 TIRFM, Olympus) was used for TIR illumination. All experiments were carried out at room temperature in medium without phenol red supplemented with an oxygen scavenger and a redox-active photoprotectant to minimize photobleaching (*Vogelsang et al., 2008*). For cell surface labeling of mXFP-gp130, antiGFP-NB$^{DY647}$ and antiGFP-NB$^{RHO11}$ were added to the medium at equal concentrations (2 nM) and incubated for at least 5 min by that ensuring >90% binding given the 0.45 nM binding affinity (*Kirchhofer et al., 2010*). The nanobodies were kept in the bulk solution during the whole experiment in order to ensure high equilibrium binding to mXFP-gp130. Dimerization of mXFP-gp130 was probed before and after incubation with either 20 nM HyIL-6 or 1 µM of the IL-6 mutants (Mut3, C7, A1, IL-6 wt). Image stacks of 150 frames were recorded at 32 ms/frame. For simultaneous dual color acquisition, antiGFP-NB$^{RHO11}$ was excited by a 561 nm diode-pumped solid-state laser at 0.95 mW (~32 W/cm$^2$) and antiGFP-NB$^{DY647}$ by a 642 nm laser diode at 0.65 mW (~22 W/cm$^2$). Fluorescence was detected using a spectral image splitter (DualView, Optical Insight) with a 640 DCXR dichroic beam splitter (Chroma) in combination with the bandpass filter 585/40 (Semrock) for detection of RHO11 and 690/70 (Chroma) for detection of DY647 dividing each emission channel into 512 × 256 pixel. In order to probe the dimerization/ligand binding of/to endogenous gp130 presented on HeLa cells, each ligand was (HyIL-6, Mut3, A1 and C7) conjugated to DY547 and DY647, respectively. Prior the experiment HeLa cells were incubated with 10 nM of both dye-conjugated ligands (DY547 and DY647) for 10 min at room temperature and dual colour experiments have been performed like described above.

Single molecule localization and single molecule tracking were carried out using the multiple-target tracing (MTT) algorithm (*Sergé et al., 2008*) as described previously (*You et al., 2016*). Step-length histograms were obtained from single molecule trajectories and fitted by two fraction mixture model of Brownian diffusion. Average diffusion constants were determined from the slope (2–10 steps) of the mean square displacement versus time lapse diagrams. Immobile molecules were identified by the density-based spatial clustering of applications with noise (DBSCAN) algorithm as described recently (*Röder et al., 2014*). For comparing diffusion properties and for co-tracking analysis, immobile particles were excluded from the data set.

Prior to co-localization analysis, imaging channels were aligned with sub-pixel precision by using a spatial transformation. To this end, a transformation matrix was calculated based on a calibration measurement with multicolor fluorescent beads (TetraSpeck microspheres 0.1 mm, Invitrogen) visible in both spectral channels (cp2tform of type 'affine', The MathWorks MATLAB 2009a).

Individual molecules detected in the both spectral channels were regarded as co-localized, if a particle was detected in both channels of a single frame within a distance threshold of 100 nm radius. For single molecule co-tracking analysis, the MTT algorithm was applied to this dataset of co-localized molecules to reconstruct co-locomotion trajectories (co-trajectories) from the identified population of co-localizations. For the co-tracking analysis, only trajectories with a minimum of 10 steps (~320 ms) were considered. The relative fraction of co-tracked molecules was determined with

respect to the absolute number of trajectories and corrected for gp130 stochastically double-labeled with the same fluorophore species as follows:

$$AB^* = \frac{AB}{2 \times \left[ \left( \frac{A}{A+B} \right) \times \left( \frac{B}{A+B} \right) \right]}$$

$$rel.\ co-locomotion = \frac{2 \times AB^*}{(A+B)}$$

where A, B, AB and AB* are the numbers of trajectories observed for Rho11, DY647, co-trajectories and corrected co-trajectories, respectively.

## Imaging of receptors in endosomes

For tracing IL-6 uptake into early endosomes, HeLa cells were transiently transfected (PEI) with XFP-gp130 and seeded on 12 mm cover slides placed in a 24-well plate. Cells were stimulated for 45 min at 37C with 20 nM of HYIL-6$^{DY547}$ or 40 nM of Mut3$^{DY547}$, C7$^{DY547}$ and A1$^{DY547}$ respectively. Cells were PFA fixed (4%,15 min in PBS), and washed 3x with PBS. Cells were permeabilized in a Methanol buffer (90% MeOH, MES, 10 mM EDTA 100 µM, MgCl$_2$100 µM) for 1 min and washed 3x with PBS. Subsequently, cells were incubated in blocking buffer (TBS + 1% BSA = TBSA) for 20 min. Cells were incubated with the primary antibody against EEA1 (mouse-anti-human, 1:200, eBioscience, #14-9114-80) in TBSA for 45 min and washed 3x with TBSA. Cells were incubated with the secondary antibody (donkey-anti-mouse$^{Alexa647}$1:200, Life Technologies, #A31571) in TBSA for 45 min, washed 3x with TBSA and mounted under coverslips using Vectashield mounting medium containing DAPI (Vector Laboratories) and viewed using an LSM 700 confocal microscope (Carl Zeiss).

## Acknowledgements

We thank members of the Moraga, Mitra and Kazemian laboratories for helpful advice and discussion. We thank Dynamic Biosensors for their help characterizing IL-6 variants binding properties. This work was supported by the StG, LS6, Wellcome-Trust-202323/Z/16/Z (IM EP), ERC-206-STG grant (IM JMF PKF), EMBO (SW 454–2017), DFG (SFB 944, P8/Z, JP), National Heart, Lung and Blood Institute (*K22*HL125593, MK) and Contrat de Plan Etat Région Hauts de France and Institut pour la Recherche sur le Cancer de Lille (SM).

## Additional information

### Funding

| Funder | Grant reference number | Author |
|---|---|---|
| Horizon 2020 Framework Programme | 714680 | Jonathan Martinez-Fabregas Paul K Fyfe Ignacio Moraga |
| EMBO | 454-2017 | Stephan Wilmes |
| National Heart, Lung, and Blood Institute | *K22*HL125593 | Majid Kazemian |
| Wellcome Trust | Sir Henry Dale Fellowship 202323/Z/16/Z | Elizabeth Pohler Ignacio Moraga |
| Royal Society | Sir Henry Dale Fellowship 202323/Z/16/Z | Elizabeth Pohler Ignacio Moraga |
| Contrat de Plan Etat Région Hauts de France and Institut pour la Recherche sur le Cancer de Lille | | Suman Mitra |
| Deutsche Forschungsgemeinschaft | SFB 944, P8/Z | Jacob Piehler |

The funders had no role in study design, data collection and interpretation, or the decision to submit the work for publication.

## Author contributions
Jonathan Martinez-Fabregas, Stephan Wilmes, Suman Mitra, Conceptualization, Investigation, Methodology; Luopin Wang, Data curation, Formal analysis, Investigation; Maximillian Hafer, Juliane Lokau, Adeline Cozzani, Methodology; Elizabeth Pohler, Investigation, Methodology; Christoph Garbers, Jacob Piehler, Conceptualization, Methodology; Paul K Fyfe, Resources; Majid Kazemian, Conceptualization, Formal analysis, Investigation, Methodology; Ignacio Moraga, Conceptualization, Funding acquisition, Investigation, Methodology, Project administration

## Author ORCIDs
Jonathan Martinez-Fabregas (iD) https://orcid.org/0000-0001-5809-065X
Stephan Wilmes (iD) https://orcid.org/0000-0002-4112-710X
Luopin Wang (iD) https://orcid.org/0000-0001-5758-4092
Maximillian Hafer (iD) https://orcid.org/0000-0003-0853-2637
Juliane Lokau (iD) https://orcid.org/0000-0002-2573-7282
Jacob Piehler (iD) https://orcid.org/0000-0002-2143-2270
Majid Kazemian (iD) https://orcid.org/0000-0001-7080-8820
Suman Mitra (iD) https://orcid.org/0000-0002-3426-371X
Ignacio Moraga (iD) https://orcid.org/0000-0001-9909-0701

## Decision letter and Author response
Decision letter https://doi.org/10.7554/eLife.49314.sa1
Author response https://doi.org/10.7554/eLife.49314.sa2

# Additional files

## Supplementary files
• Supplementary file 1. Number of STAT3 binding sites.

• Transparent reporting form

## Data availability
Sequencing data have been deposited in GEO under accession number code GSE130810.

The following dataset was generated:

| Author(s) | Year | Dataset title | Dataset URL | Database and Identifier |
|---|---|---|---|---|
| Martinez-Fabregas J, Wilmes S, Wang L, Hafer M, Pohler E, Lokau J, Garbers C, Cozzani A, Piehler J, Kazemian M, Mitra S, Moraga I | 2019 | Kinetics of cytokine receptor trafficking determine signaling and functional selectivity | https://www.ncbi.nlm.nih.gov/geo/query/acc.cgi?acc=GSE130810 | NCBI Gene Expression Omnibus, GSE130810 |

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
