## [Decision Letter]

**Acceptance summary:**

This manuscript by Martinez-Fabregas et al. provides interesting evidence for a role of ligand affinity-dependent regulation of receptor trafficking leading to modification of intracellular signalling engagements and outcomes, using the IL6-IL6R system. It addresses a biologically important and technically interesting question of how binding affinity between cytokine ligands and receptors can differentially modulate signalling output. The case with cytokine receptors is contrasted with the biased signalling known in GPCRs, where different ligands produce different responses from the same receptor via allosteric modulations. To study this system, the authors generated a panel of IL-6 variants with differing binding affinities for gp130. They find that these ligands trigger biased signalling based on their affinities and use this system to extract mechanistic insights. More broadly, the authors suggest these results highlight that manipulation of cytokine-receptor binding parameters could be a useful strategy to decouple cytokine functional pleiotropy, which is a source of toxicity in cytokine-based therapies. This is a very impressive piece of work and an important contribution to the field.

**Decision letter after peer review:**

Thank you for sending your article entitled "Kinetics of cytokine receptor trafficking determine signaling and functional selectivity" for peer review at *eLife*. Your article is being evaluated by two peer reviewers, one of whom is a member of our Board of Reviewing Editors, and the evaluation is being overseen by Satyajit Rath as the Senior Editor.

Given the list of essential revisions, including new experiments, the editors and reviewers invite you to respond within the next two weeks with an action plan and timetable for the completion of the additional work. We plan to share your responses with the reviewers and then issue a binding recommendation.

Both reviewers have evaluated your manuscript and find that your work addresses a biologically important and technically interesting question of how binding affinity between cytokine ligands and receptors can differentially modulate signaling output. However, both feel that the mechanism that you propose is not clearly described and alternative possibilities are not convincingly excluded. The important points to address are: 1) the data suggests a kinetic effect rather than an affinity effect, whereas an affinity effect is proposed, and 2) the alternative explanation that the different affinity ligands bind in different configurations, leading to differential receptor clustering and thereby different signalling output should be disproven. Some of the imaging data or experiments that affect trafficking are not clear. The labelled ligand experiments with confocal microscopy should be performed as time lapse as function of different ligand concentrations to convincingly demonstrate the differential trafficking of the high/low affinity ligands. In the PITSTOP experiment no statistical significance is given and only again the high affinity HylL6 is showing a clear effect. Are there alternative ways (genetic?) to affect the trafficking that yield more convincing results. In addition, please address point-by-point the comments of the individual reviews.

Reviewer #1:

This study claims that ligand-receptor lifetimes (investigated by generating iL6 mutants with different binding affinities) affect cytokine signalling trough differential trafficking of multimeric receptors comprising gp130. IL-6 stimulates signaling via a hexameric receptor comprised of two copies of IL-6, two gp130 and two IL-6Rα receptor subunits. The authors used yeast surface display to engineer a series of IL-6 variants binding gp130 with different affinities. Three variants with different apparent affinities for the receptor complex were investigated: A1 (648 nM) and the high affinity C7 (6.2 nM) and Mut3 (379 pM). My initial enthusiasm for this study was hampered by inconsistencies in some of the interpretation of the experiments. The major point that the authors make is that low affine binding of ligand to the receptor complex impedes receptor internalization leading to STAT3 biased responses at the plasma membrane whereas high affine binding to gp130 causes its internalization also enabling the activation of STAT1 on endosomes. The difference in the ratio of STAT1/STAT3 activity causes differential gene expression. The authors explain this difference in trafficking by the difference in the receptor-ligand lifetime (effectively the difference in koff). In the first result chapter a minor inconsistency confused me: at the end of the paragraph "The binding affinities of three of those variants, belonging to the first and second libraries, were confirmed by surface plasmon resonance (SPR) studies, with values ranging from 648 nM (A1) to 6.2 nM (C7) and 379 pM (Mut3) (Figure 1G and Figure 1—figure supplement 1C). The table in Figure 1G confirms that both Mut3 and C7 are high affinity, low nM Kd ligands with a KD ratio of approximately 16 (6nM/0.376nM). However, in Figure 1E where gp130 binding is measured the affinities seem to differ by two orders of magnitude for C7 and Mut3, whereas A1 does not seem to bind at all (despite a having a similar affinity as IL-6). What is the binding affinity of HylL-6 for gp130? In general, several further inconsistencies weaken the conclusions drawn in this manuscript. For example, the high affinity C7 mutant ligand is two orders of magnitude less potent in activating STAT1/STAT3 as the high affinity ligand HyIL-6 (Figure 2A,B). Even for the sub-nano-molar Mut3 mutant (super)-high affinity binder, its potency is one order of magnitude less in activating STAT1/STAT3 as the high affinity ligand HyIL-6. Why is this so? Is the affinity of HylL-6 even an order of magnitude higher as that of Mut3? Is a possible explanation that the different ligands induce a different receptor complex configuration by changing the interactions between extracellular domains? Could the balance between IL-6Rα engagement versus gp130 engagement affect signaling outcome? This is indicated by the fact that IL-6 wild type (wt), requires IL-6Rα expression to activate signalling whereas the high affinity Hyper IL-6 (HyIL-6) only requires gp130 and potently triggers signaling in cells lacking IL-6Rα. This all points at a possible other mechanism, where the different mutant ligands generate different extracellular conformations of the gp130 that could engage in different heterotypic receptor clusters with different signalling capacities.

The imaging data that should support the hypothesis that the lifetime of the receptor-ligand complexes affects the trafficking are not clear. For example, the authors detected the high affinity Mut3 ligand in intracellular compartments that partially co-localized with EEA1 early endosome marker, whereas no fluorescence was detected for the high affinity C7 mutant ligand (factor 3 difference in off rate) (Figure 3D,E). Should this not give some internalisation for high affinity ligands? For any ligand at least binding at the plasma membrane should be visible for concentrations higher than the Kd of the ligand/receptor complex, which is not visible for C7 (40 nM ligand concentration versus 6 nM Kd; Figure 3D). The off-rates given in table one let us approximate the lifetime of the complexes which for the C7 and Mut3 is in the order of ~20 min and ~60 min, which is very long in comparison with the STAT signalling time-scale. The authors also state: "We detected significant ligand induced gp130 co-locomotion for all IL-6 variants except for A1, with levels of co-locomotion paralleling gp130 binding affinities, i.e. HyIL-6>Mut3>C7>A1. Should A1 not have similar to WT IL-6 some engagement of GP130? Is this also not dependent on the outside concentration of A1 (saturating concentrations)? In general, the co-locomotion of receptors for any of the mutant ligands is very sparse (Figure 3—figure supplement 2). What is the reason for this if the functional receptor is a hexameric complex? When the authors compare gp130 degradation, only the high affinity ligand HyIL-6 causes significant degradation. Is it not so that if Mut3 is the high affinity ligand that was actually found on endosomes (Figure 3D) that it should also end up in lysosomes by the usual Rab5/Rab7 switch for early to late endosomes? In the PITSTOP experiment no statistical significance is given and only again the high affinity HylL6 is showing a clear effect. However, this effect on receptor degradation is similar to the non-stimulated case, making the whole link to ligand-receptor lifetime regulating trafficking weaker. The idea of competitive binding for the STATs to gp130 should also be true for gp130 on intracellular compartments? This idea could also be further tested by enhancing the relative expression of STAT1 (instead of knock down of STAT3) versus STAT3 in cells. Is this effect not due to a differential regulation of STAT-binding phosphorylation sites on gp130 by intra-cellular protein tyrosine phosphatases such as PTPN1/2? The factor signalling time is also not taken into consideration in the down-stream differential gene expression effect. Despite the interesting results that are presented, I feel that at this stage the authors have to do substantial additional work to prove that ligand-receptor lifetimes affect receptor trafficking and thereby multimeric receptor signalling via STAT1/STAT3.

Reviewer #2:

This manuscript by Martinez-Fabregas et al. addresses a biologically important and technically interesting question of how binding affinity between cytokine ligands and receptors can differentially modulate signaling output. The authors nicely frame the biological background of this problem in the Introduction. The case with cytokine receptors is contrasted with the biased signaling known in GPCRs, where different ligands produce different responses from the same receptor via allosteric modulations. The author's point out that this cannot be applied to cytokine receptors with a single membrane-spanning domain (although I question their certainty on this, see minor points).

To study this system, the authors generated a panel of IL-6 variants with differing binding affinities for gp130. They find that these ligands trigger biased signaling based on their affinities and use this system to extract mechanistic insights. More broadly, the authors suggest these results highlight that manipulation of cytokine-receptor binding parameters could be a useful strategy to decouple cytokine functional pleiotropy, which is a source of toxicity in cytokine-based therapies. Overall I think it's a very impressive piece of work and an important contribution to the field. I list a couple of more detailed points and questions below:

• In their study of differential STAT3/STAT1 activation the authors note that the different ligands drove differential phosphorylation amplitudes, which could not be rescued by increases in ligand concentration. This observation suggests that it is not affinity to which the receptors respond, but rather must be kinetic off-rate. If increasing ligand concentration cannot rescue the signaling differences, then receptor occupancy is not what the cell is measuring. The only other parameter is binding dwell time-very much the way T cell receptor distinguishes peptide MHC ligands. I suggest that the author's data does not support binding affinity as the differentiator, but rather binding dwell time. In the language they use in the text, this distinction is important and it would be useful if the authors elaborated on it more explicitly.

• The next section, "short lived IL-6-gp130 complexes fail to traffic to intracellular compartments", does essentially claim a kinetic discrimination mechanism-although the authors don't directly say this. The language still uses terms like "binding energy". I think the authors should take a more clear stand on kinetic discrimination vs. binding strength. Statements such as "one of the first steps affected by changes in gp130 binding affinity would be the assembly kinetics" are rather unclear. Why would binding affinity affect this? Binding affinity only affects receptor occupancy, and should always be compensated by changing ligand concentration. If the data indicate ligand concentration cannot compensate, then a kinetic proof reading mechanism in which short dwelling events fail to activate-no matter how many there are-seems likely, if not necessary. The nuance here is that kinetic proof reading requires an out-of-equilibrium process and rate-limiting kinetic intermediates (see, for example, Proc. Natl. Acad. Sci. USA 2016, 113, 29: 8218 and Science 2019, 363: 1098). Is the gp130 system capable of kinetic proof reading based on what is known? Perhaps internalization breaks equilibrium and provides the rate-limiting step? It’s not obvious to me that it is, but being able to say anything about this based on their data would be extremely interesting.

• In the Discussion, the author's seem to dismiss kinetic proof reading as unlikely or difficult to reconcile. However, isn't the STAT competition model the authors articulate providing a form of kinetic proof reading? To some degree this is only a matter of semantics in the definition of the term, kinetic proof reading. But I suggest the fact that the mechanism the authors have identified achieves a form of kinetic discrimination is an important realization, and probably should be identified as a form of kinetic proof reading.

---

## [Author Response]

Both reviewers have evaluated your manuscript and find that your work addresses a biologically important and technically interesting question of how binding affinity between cytokine ligands and receptors can differentially modulate signaling output. However, both feel that the mechanism that you propose is not clearly described and alternative possibilities are not convincingly excluded.The important points to address are: 1) the data suggests a kinetic effect rather than an affinity effect, whereas an affinity effect is proposed.

We fully agree with the reviewers that our data strongly argues in favor of a kinetic effect rather than affinity, and we use terms like complex half-life and kinetics proof-reading throughout the manuscript to convey this. However, we see now that message did not come across in the manuscript clearly enough and we apologize for any confusion resulting from our language. We have amended the manuscript to clearly state this fact in the revised version. We provide some further points regarding this in the point-by-point answers to the reviewers (below).

2) the alternative explanation that the different affinity ligands bind in different configurations, leading to differential receptor clustering and thereby different signalling output should be disproven.

Despite our best efforts we have not been able to obtain protein crystals of our mutants in complex with gp130 and we can, therefore, not unequivocally exclude changes in binding topology. However, while it is possible that the new IL-6 variants bind gp130 in alternative conformations, we believe this to be highly improbable for the following reasons:

a) When generating the new IL-6 variants we kept the backbone of the cytokine intact and introduced mutations only in the cytokine-receptor binding interface. There is a large body of work done by the Garcia lab and others showing that introducing mutations in cytokine-receptor binding interfaces is a valid strategy to fine-tune binding affinities without changing topology (Levin AM, Nature, 2012; Juntilla IS, Nat.Chem.Biol., 2012; Moraga et al., 2015; Mendoza et al., 2017). In fact, to our knowledge there is not a single example in the literature of a mutation in a cytokine binding interface that alters the overall binding architecture of the ligand-receptor complex, strongly suggesting that the evolutionary pressure that maintains the canonical architecture of the cytokine-receptor complex is too high to be overcome by introducing few mutations in the binding interface. Indeed, the only ligands reported to alter canonical binding architectures are surrogate ligands such as diabodies and peptides (Livnah et al., 1998; Moraga et al., 2015; Mohan K, Science, 2019).

b) We have additional data, which we can include upon request, showing that IL-6 variants still rely on a classical site-2/site-3 interaction to recruit gp130 and signal, strongly supporting the hypothesis that our mutants recruit gp130 in a similar manner to wild type IL-6. We have generated a dominant-negative version of Mut3, by introducing mutations that disrupt its binding to gp130 via site-3. As a consequence, this new Mut3 variant behaves as an antagonist, able to bind gp130 with high affinity via site-2, but not able to trigger signaling due to its inability to recruit a second molecule of gp130 via site-3. These data confirm that Mut3 uses a site-2/site-3 architecture to assemble a signaling-compatible gp130 complex.

c) We have performed new TIRF microscopy experiments to confirm the stoichiometry of the complex formed by the IL-6 mutants. These experiments have shown that all IL-6 mutants, as well as HyIL-6, recruit two molecules of gp130 to trigger signaling, excluding the possibility of larger clusters of receptors induced by the mutants. This new data are now part of a new Figure 3 and Figure 3—figure supplements 1 and 2.

We believe that we have strong experimental evidences that support a canonical gp130 binding architecture by the different IL-6 mutants. However, in the absence of a crystal structure we agree that we cannot formally disprove this possibility and we have reflected this in the Discussion.

Some of the imaging data or experiments that affect trafficking are not clear. The labelled ligand experiments with confocal microscopy should be performed as time lapse as function of different ligand concentrations to convincingly demonstrate the differential trafficking of the high/low affinity ligands.

We agree with the reviewers that this part of the manuscript needs to be strengthened. We have now developed a flow cytometry-based method to monitor IL-6 internalization at a wide range of time and ligand concentrations. These new data sets, presented now in new Figure 3F-G, further reinforce our initial observations that short-lived IL-6-gp130 complexes fail to traffic to intracellular compartments and to induce gp130 degradation.

In the PITSTOP experiment no statistical significance is given and only again the high affinity HylL6 is showing a clear effect.

Blocking receptor internalization is very challenging. High doses of inhibitors like Pitstop will stress the cells leading to overall poor signaling. Complete silencing of clathrin or the use of dynamin dominant negative mutants produces similar outcome. Based on this, we had to find a compromise between inhibition of traffic and cellular health, which led to variability in the experimental outcome. However, we obtained similar effects on signaling when using Pitstop or clathrin silencing, which we interpret as a solid evidence that gp130 traffic was required for STAT1 activation but not STAT3 activation. Nevertheless, we understand the reviewer's concerns and we have now included new data as new Figure 4 A-B, which more clearly show that IL-6 variants exhibit a significant defect in inducing gp130 degradation as compared to HyIL-6. Additionally, using fluorescently labeled HyIL-6, we have now more accurately quantified the inhibition of gp130 internalization upon Pitstop treatment (new Figure 4C). Indeed, Pitstop treatment elicits a very strong inhibition of HyIL-6 internalization. Below, we expand on why only HyIL-6 induces degradation of gp130 in the answers to the reviewer 1 questions.

Overall, we believe that the new modifications that we have introduced in the text to better explain our conclusions, together with the large array of new experimental data that we are providing, fully support our initial hypotheses and address the majority of concerns raised by the editors and reviewers.

Reviewer #1:This study claims that ligand-receptor lifetimes (investigated by generating iL6 mutants with different binding affinities) affect cytokine signalling trough differential trafficking of multimeric receptors comprising gp130. IL-6 stimulates signaling via a hexameric receptor comprised of two copies of IL-6, two gp130 and two IL-6Rα receptor subunits. The authors used yeast surface display to engineer a series of IL-6 variants binding gp130 with different affinities. Three variants with different apparent affinities for the receptor complex were investigated: A1 (648 nM) and the high affinity C7 (6.2 nM) and Mut3 (379 pM). My initial enthusiasm for this study was hampered by inconsistencies in some of the interpretation of the experiments. The major point that the authors make is that low affine binding of ligand to the receptor complex impedes receptor internalization leading to STAT3 biased responses at the plasma membrane whereas high affine binding to gp130 causes its internalization also enabling the activation of STAT1 on endosomes. The difference in the ratio of STAT1/STAT3 activity causes differential gene expression. The authors explain this difference in trafficking by the difference in the receptor-ligand lifetime (effectively the difference in koff). In the first result chapter a minor inconsistency confused me: at the end of the paragraph "The binding affinities of three of those variants, belonging to the first and second libraries, were confirmed by surface plasmon resonance (SPR) studies, with values ranging from 648 nM (A1) to 6.2 nM (C7) and 379 pM (Mut3) (Figure 1G and Figure 1—figure supplement 1C). The table in Figure 1G confirms that both Mut3 and C7 are high affinity, low nM Kd ligands with a KD ratio of approximately 16 (6nM/0.376nM). However, in Figure 1E where gp130 binding is measured the affinities seem to differ by two orders of magnitude for C7 and Mut3, whereas A1 does not seem to bind at all (despite a having a similar affinity as IL-6).

We appreciate the reviewer's comments and apologize for any confusion resulting from the lack of clarity in our presentation. The binding curves presented in Figure 1E correspond to the on-yeast binding titrations. Although this experiment is very informative and useful to stratify the different mutants based on their relative binding affinities for further studies, it does not provide a K_D_ binding constant as the one obtained in the SPR measurements. Additionally, it cannot detect binding of ligands with relatively low binding affinities, as A1 and IL-6 wt, explaining why we do not get binding with these variants in these experimental set up. Therefore, it is not surprising to see differences in binding values from the two different set of experiments, with the SPR providing the most accurate measurement (Moraga et al., 2015). We have now amended the text (Paragraph three of the Results section) to clarify this point.

What is the binding affinity of HylL-6 for gp130? In general, several further inconsistencies weaken the conclusions drawn in this manuscript. For example, the high affinity C7 mutant ligand is two orders of magnitude less potent in activating STAT1/STAT3 as the high affinity ligand HyIL-6 (Figure 2A,B). Even for the sub-nano-molar Mut3 mutant (super)-high affinity binder, its potency is one order of magnitude less in activating STAT1/STAT3 as the high affinity ligand HyIL-6. Why is this so? Is the affinity of HylL-6 even an order of magnitude higher as that of Mut3?

We apologize for the missing information on how IL-6 assembles its surface receptor to trigger signaling, which we believe has led to the confusion. We appreciate the point the reviewer is trying to make and have now clarified our language in the revised manuscript (Paragraph three of the Results section). Additionally, we are now providing schematic diagrams displaying the IL-6 receptor assembly kinetics modes used by the different IL-6 ligands described in this study as a new Figure 1A in the revised manuscript.

Briefly, IL-6 in a first step binds IL-6Rα receptor with high affinity. In a second step, this binary complex recruits gp130, via site-2 on the ligand-receptor interface, to form a hetero-trimeric complex, which cannot trigger signaling. In a third step, two hetero-trimeric complexes come together, via site-3 interaction, to finally form the signaling-compatible hexameric complex. HyIL-6, which is a fusion protein comprised of IL-6Rα and IL-6, directly binds gp130 on site-2 and then recruits a second gp130 molecule via site-3 to trigger signaling. It is important to emphasize that the presence of IL-6Rα helps to stabilize the complex by contributing to the formation of the site-2 and site-3 binding interfaces (new Figure 1—figure supplement 1A). Therefore, in the absence of IL-6Rα, IL-6 wildtype cannot bind gp130 and trigger signaling.

To produce our IL-6 variants we have mutated the site-2 binding interface on IL-6, with all of the mutants exhibiting an unaltered wild-type site-3 interface. Consequently, the IL-6 variants binding affinities reported in the manuscript correspond exclusively to binding of the mutants to site-2 and not the overall affinity of the complex. What this means is that we have improved the ability of the mutants to find gp130 on the cell surface and therefore their efficiency in forming the intermediate IL-6/gp130 (site-2) heterodimeric complex in the absence of IL-6Rα. However, in order to trigger signaling this heterodimeric (mutant/gp130) complex still requires the recruitment of a second molecule of gp130 via site-3, which exhibit very low affinity, ultimately defining the final half-life of the complex formed. HyIL-6, which comprises IL-6Rα, forms a more stable site-3 binding interface, by stabilization of this binding interface via IL-6Rα/gp130 contacts, therefore forming a longer-lived complex than our engineered mutants, explaining its stronger overall potency.

We have experimental data that support this binding sequence for the engineered variants, using a Mut3 dominant negative form and can provided upon request. We have ablated site-3 binding on Mut3 by replacing W157, which is located in the center of this binding interface, with a cysteine residue, allowing us to place a FITC molecule via a click chemistry reaction. This in turn completely ablates the ability of Mut3 to recruit a second gp130 molecule via its site-3 interface to trigger signaling. Importantly, Mut3DN inhibits signaling by the lower affinity C7 mutant proving that it is still binding with high affinity to gp130 via its site-2 interface. However, Mut3DN cannot inhibit signaling induced by HyIL-6, even at 50-fold excess, strongly supporting a more stable complex formed by HyIL-6.

Is a possible explanation that the different ligands induce a different receptor complex configuration by changing the interactions between extracellular domains?

Although possible, we believe this possibility to be very unlikely. Please see the answer that we provide above.

Could the balance between IL-6Rα engagement versus gp130 engagement affect signaling outcome? This is indicated by the fact that IL-6 wild type (wt), requires IL-6Rα expression to activate signalling whereas the high affinity Hyper IL-6 (HyIL-6) only requires gp130 and potently triggers signaling in cells lacking IL-6Rα. This all points at a possible other mechanism, where the different mutant ligands generate different extracellular conformations of the gp130 that could engage in different heterotypic receptor clusters with different signalling capacities.

As described above IL-6Rα contributes to stabilize the IL-6/receptor complex by interacting with gp130 via site-2 and site-3 binding interfaces. Thus, in the absence of IL-6Rα, the wildtype cytokine cannot recruit gp130 due to its low affinity. HyIL6, which already has IL-6Rα as part of its core, can recruit gp130 and trigger signaling. By contrast, our mutants can bind gp130 via site-2 in the absence of IL-6Rα. While gp130 dimerization is mediated by interactions via site-3 (which does not differ between the different variants), the overall lifetime of the signaling complex for our engineered IL-6 (two IL-6 with two gp130 molecules) is probably limited by the site-2 interaction. We therefore expect differential complex stabilities which ultimately impact their ability to activate STAT1 and STAT3 proteins. We want to emphasize at this point that the signaling as well as the imaging studies presented in original Figure 2,3, 4 and 5 in the manuscript were done in cells lacking IL-6Ra expression in order to characterize the real contribution of gp130 binding to signaling output, therefore explaining the stronger potency exhibited by HyIL-6 and the lack of signaling by wild-type IL-6. To address the alternative scenario suggested by the reviewer we have now performed further TIRF microscopy assays. We now provide strong evidences that all mutants form a stoichiometrically identical complex as the one formed by IL-6 wildtype and HyIL-6, which argues against the formation of larger heterotypic complexes by these mutants (new Figure 3E and Figure 3—figure supplement 2D).

The imaging data that should support the hypothesis that the lifetime of the receptor-ligand complexes affects the trafficking are not clear. For example, the authors detected the high affinity Mut3 ligand in intracellular compartments that partially co-localized with EEA1 early endosome marker, whereas no fluorescence was detected for the high affinity C7 mutant ligand (factor 3 difference in off rate) (Figure 3D,E). Should this not give some internalisation for high affinity ligands? For any ligand at least binding at the plasma membrane should be visible for concentrations higher than the Kd of the ligand/receptor complex, which is not visible for C7 (40 nM ligand concentration versus 6 nM Kd; Figure 3D). The off-rates given in table one let us approximate the lifetime of the complexes which for the C7 and Mut3 is in the order of ~20 min and ~60 min, which is very long in comparison with the STAT signalling time-scale.

The affinity of the mutants for gp130 exclusively represents their binding to site-2 and not the overall stability of the complex formed. So, while the C7 and Mut3 mutants, as highlighted by the reviewer, will remain bound to gp130 for ~20 min and ~60 min, this is not true for the signaling-compatible tetrameric complex of two IL-6 and two gp130 molecules formed by these mutants, which is limited by the affinity of the mutants for site-3. As a result, even Mut3, which binds with extremely high affinity to site-2, induces a very poor internalization when compared to hyIL-6, which exhibits high binding affinity for both site-2 and site-3 binding interfaces. This said, we agree with the reviewer in that the internalization experiment should be done at a wider range of ligand concentrations and stimulation times to more accurately assess gp130 internalization by the different IL-6 mutants. To address this, we have utilized a flow-cytometry based assay. We have incubated HeLa cells with a wide range of concentrations of the different fluorescently labelled IL-6 variants, or we saturated concentrations at different time points. We have used flow cytometry to quantify the fluorescence signals resulting from intracellular fluorescence molecules. We now show in new Figure 3F and G and 4C that HyIL-6 is the only IL-6 variant that it is robustly internalized in both experimental set ups, with the other IL-6 variants exhibiting very poor internalization, thus supporting our initial observations.

The authors also state: "We detected significant ligand induced gp130 co-locomotion for all IL-6 variants except for A1, with levels of co-locomotion paralleling gp130 binding affinities, i.e. HyIL-6>Mut3>C7>A1. Should A1 not have similar to WT IL-6 some engagement of GP130? Is this also not dependent on the outside concentration of A1 (saturating concentrations)? In general, the co-locomotion of receptors for any of the mutant ligands is very sparse (Figure 3—figure supplement 2). What is the reason for this if the functional receptor is a hexameric complex?

We agree with the reviewer in that the co-locomotion, although reproducible and significant, is sparse. The reason for that is that due to ubiquitous expression of endogenous gp130 in all cell lines, which affect TIRF measurements, we used an available gp130 KO HEK293 clone obtained by CRISPR technology (Schwerd T, J. Ex. Med., 2017). Unfortunately, HEK293 cells are very poor cells for imaging, therefore producing sparse signal as detected here. We have now generated new RPE1 gp130 KO clones, which we have used to repeat the TIRF studies. We are now providing new data shown in new Figure 3B and Figure 3—figure supplement 1 and 2, that robustly show dimerization efficiencies induced by the different IL-6 ligands. Additionally, we have performed new stoichiometric analysis to show that the IL-6 variants are not inducing different receptor oligomerization on the cell surface (new Figure 3E). It is notable that RPE1 cells do not express IL-6Rα, therefore we are not measuring hexameric complex formation, but tetrameric complex (two IL-6 with two gp130) for the different IL-6 variants and IL-6 wild-type, with the exception of HyIL-6, which already has IL-6Rα. This explains why we detect very low dimerization of gp130 in response to IL-6 wt or A1 mutant.

When the authors compare gp130 degradation, only the high affinity ligand HyIL-6 causes significant degradation. Is it not so that if Mut3 is the high affinity ligand that was actually found on endosomes (Figure 3D) that it should also end up in lysosomes by the usual Rab5/Rab7 switch for early to late endosomes? In the PITSTOP experiment no statistical significance is given and only again the high affinity HylL6 is showing a clear effect. However, this effect on receptor degradation is similar to the non-stimulated case, making the whole link to ligand-receptor lifetime regulating trafficking weaker.

HyIL-6 forms a more stable complex with gp130 than the IL-6 mutants due to its stronger interaction via site-3. The different mutants, due to their weak site-3 interaction, are unable to induce strong receptor internalization and therefore degradation. Although it is true that we detect some Mut3 molecules in endosomal compartments, the levels detected are much weaker than those induced by HyIL-6, and are insufficient to produce a significant decrease in the total levels of gp130. We have now performed more detailed kinetics of gp130 degradation (new Figure 4A and B) by the different IL-6 ligands to strengthen our initial observations. Our new data clearly show that HyIL-6 induces a significantly stronger degradation than the other IL-6 ligands, followed by the Mut3 mutant. C7, A1 and IL-6 ligands do not induce gp130 degradation, but lead to some stabilization of the receptor.

The idea of competitive binding for the STATs to gp130 should also be true for gp130 on intracellular compartments? This idea could also be further tested by enhancing the relative expression of STAT1 (instead of knock down of STAT3) versus STAT3 in cells.

We thank the reviewer for their suggestion. We now provide new data (new Figure 5F) where we show the effect of STAT1 overexpression in STAT3 phosphorylation by hyIL-6. Indeed, over expression of STAT1 results in decreased STAT3 phosphorylation induced by all IL-6 variants tested, further supporting our STAT competition model.

Is this effect not due to a differential regulation of STAT-binding phosphorylation sites on gp130 by intra-cellular protein tyrosine phosphatases such as PTPN1/2? The factor signalling time is also not taken into consideration in the down-stream differential gene expression effect. Despite the interesting results that are presented, I feel that at this stage he authors have to do substantial additional work to prove that ligand-receptor lifetimes affect receptor trafficking and thereby multimeric receptor signalling via STAT1/STAT3.

Although this is a possibility, we think that it is very unlikely. It has been previously reported that STAT1 and STAT3 compete for the same phospho-Tyr on gp130 (Gerhartz et al., 1996; Schmitz et al., 2000). However, we understand the reviewer point and now provide new data (new Figure 5C-D) where we have studied the contribution of each Tyr on the gp130 ICD to STAT1 and STAT3 phosphorylation upon HyIL-6 treatment. Our data clearly show that STAT1 and STAT3 compete for the same Tyr in gp130, and only after mutating all four Tyr to Phe we were able to block STAT1/STAT3 activation by HyIL-6.

Reviewer #2:This manuscript by Martinez-Fabregas et al. addresses a biologically important and technically interesting question of how binding affinity between cytokine ligands and receptors can differentially modulate signaling output. The authors nicely frame the biological background of this problem in the Introduction. The case with cytokine receptors is contrasted with the biased signaling known in GPCRs, where different ligands produce different responses from the same receptor via allosteric modulations. The author's point out that this cannot be applied to cytokine receptors with a single membrane-spanning domain (although I question their certainty on this, see minor points).To study this system, the authors generated a panel of IL-6 variants with differing binding affinities for gp130. They find that these ligands trigger biased signaling based on their affinities and use this system to extract mechanistic insights. More broadly, the authors suggest these results highlight that manipulation of cytokine-receptor binding parameters could be a useful strategy to decouple cytokine functional pleiotropy, which is a source of toxicity in cytokine-based therapies. Overall I think it's a very impressive piece of work and an important contribution to the field. I list a couple of more detailed points and questions below:• In their study of differential STAT3/STAT1 activation the authors note that the different ligands drove differential phosphorylation amplitudes, which could not be rescued by increases in ligand concentration. This observation suggests that it is not affinity to which the receptors respond, but rather must be kinetic off-rate. If increasing ligand concentration cannot rescue the signaling differences, then receptor occupancy is not what the cell is measuring. The only other parameter is binding dwell time-very much the way T cell receptor distinguishes peptide MHC ligands. I suggest that the author's data does not support binding affinity as the differentiator, but rather binding dwell time. In the language they use in the text, this distinction is important and it would be useful if the authors elaborated on it more explicitly.

We fully agree with the reviewer. We have amended the text to explicitly state that IL-6/gp130 binding dwell time is the major factor contributing to biased signaling in the gp130 system.

• The next section, "short lived IL-6-gp130 complexes fail to traffic to intracellular compartments", does essentially claim a kinetic discrimination mechanism-although the authors don't directly say this. The language still uses terms like "binding energy". I think the authors should take a more clear stand on kinetic discrimination vs. binding strength. Statements such as "one of the first steps affected by changes in gp130 binding affinity would be the assembly kinetics" are rather unclear. Why would binding affinity affect this? Binding affinity only affects receptor occupancy, and should always be compensated by changing ligand concentration. If the data indicate ligand concentration cannot compensate, then a kinetic proof reading mechanism in which short dwelling events fail to activate-no matter how many there are-seems likely, if not necessary. The nuance here is that kinetic proof reading requires an out-of-equilibrium process and rate-limiting kinetic intermediates (see, for example, Proc. Natl. Acad. Sci. USA 2016, 113, 29: 8218 and Science 2019, 363: 1098). Is the gp130 system capable of kinetic proof reading based on what is known? Perhaps internalization breaks equilibrium and provides the rate-limiting step? It’s not obvious to me that it is, but being able to say anything about this based on their data would be extremely interesting.

We thank the reviewer for their suggestions. We agree that the most likely scenario that explains our results is a kinetics discrimination mechanism. Indeed, we had stated in the Discussion that “These data suggest that non-detectable short-lived IL-6/gp130 complexes can partially engage signaling, but fail to trigger a full response, evoking a kinetic-proof reading model”. However, we agree with the reviewer that the terminology should be more clear and possible mechanisms need to be discussed. We have amended the language in the Discussion to better define these concepts and more clearly state our conclusions. Regarding the reviewer's comments stating the need of an out-of-equilibrium process and rate-limiting kinetics intermediates for a kinetic-proof reading model, we believe that the gp130 system exhibits those requisites. As pointed out by the reviewer, receptor internalization is the critical event that breaks the equilibrium by irreversibly capturing intact signaling complexes into endosomes. We have recently quantitatively described this effect in a systematic study with engineered IL-13 variants (Moraga et al., 2015), showing the critical role of interaction rate constants in such a non-equilibrium process. For the engineered gp130 agonists, we clearly observed that endocytosis is altered, which presumably is caused by differential stability of ligand-receptor interaction (i.e. the off-rate). We have followed the reviewer’s advice and introduce a new chapter in the Discussion to describe these concepts (Discussion paragraph four).

• In the Discussion, the author's seem to dismiss kinetic proof reading as unlikely or difficult to reconcile. However, isn't the STAT competition model the authors articulate providing a form of kinetic proof reading? To some degree this is only a matter of semantics in the definition of the term, kinetic proof reading. But I suggest the fact that the mechanism the authors have identified achieves a form of kinetic discrimination is an important realization, and probably should be identified as a form of kinetic proof reading.

We fully agree with the idea that cytokine receptor complexes use a kinetic proof reading mechanism to define their signaling output and biological responses. We initially decided to be cautious in our use of the term as there might be some nuances between the cytokine system and other systems (e.g. TCR system) where kinetic proof reading has been proposed. However, we agree that our STAT competition model could be englobed within a more general kinetic proof reading mechanism. We have amended the Discussion to state this more clearly.